# The hidden structure of human enamel

Elia Beniash [1,2,3,4,9], Cayla A. Stifler[5,9], Chang-Yu Sun[5], Gang Seob Jung [6,7], Zhao Qin[6,7], Markus J. Buehler[6,7] & Pupa U.P.A. Gilbert [5,8]

Enamel is the hardest and most resilient tissue in the human body. Enamel includes morphologically aligned, parallel, ~50 nm wide, microns-long nanocrystals, bundled either into 5-μm-wide rods or their space-filling interrod. The orientation of enamel crystals, however, is poorly understood. Here we show that the crystalline c-axes are homogenously oriented in interrod crystals across most of the enamel layer thickness. Within each rod crystals are not co-oriented with one another or with the long axis of the rod, as previously assumed: the c-axes of adjacent nanocrystals are most frequently mis-oriented by 1°–30°, and this orientation within each rod gradually changes, with an overall angle spread that is never zero, but varies between 30°–90° within one rod. Molecular dynamics simulations demonstrate that the observed mis-orientations of adjacent crystals induce crack deflection. This toughening mechanism contributes to the unique resilience of enamel, which lasts a lifetime under extreme physical and chemical challenges.

[1] Department of Oral Biology, School of Dental Medicine, UPitt, Pittsburgh, PA 15261, USA. [2] Department of Bioengineering, Swanson School of Engineering, University of Pittsburgh, Pittsburgh, PA 15261, USA. [3] Center for Craniofacial Regeneration, School of Dental Medicine, UPitt, Pittsburgh, PA 15261, USA. [4] McGowan Institute for Regenerative Medicine, School of Dental Medicine, UPitt, Pittsburgh, PA 15261, USA. [5] Department of Physics, UW-Madison, Madison, WI 53706, USA. [6] Department of Civil and Environmental Engineering, Massachusetts Institute of Technology, Cambridge, MA 02139-4307, USA. [7] Laboratory for Atomistic & Molecular Mechanics, Massachusetts Institute of Technology, Cambridge, MA 02139-4307, USA. [8] Departments of Chemistry, Geoscience, Materials Science and Engineering, UW-Madison, Madison, WI 53706, USA. [9] These authors contributed equally: Elia Beniash, Cayla A. Stifler. Previously publishing as Gelsomina De Stasio: Pupa U. P. A. Gilbert. Correspondence and requests for materials should be addressed to M.J.B. (email: mbuehler@mit.edu) or to P.U.P.A.G. (email: pupa@physics.wisc.edu)

Dental enamel is the most highly mineralized tissue in the human body. Its outstanding mechanical properties combine the extreme hardness and stiffness with exceptional resilience, which enables it to withstand hundreds of masticatory cycles with biting forces of up to 770 N[1], in the harsh environment of the oral cavity, which also undergoes extreme pH and temperature fluctuations within the human body. Despite the fact that it does not remodel or repair, it lasts decades without catastrophic failure[2–4]. It covers the tooth crowns in humans[5,6] and in all tetrapods[7], and enamel-like tooth coatings first appeared 500 million years ago in conodonts[8,9]. It is composed of a hard mineral, carbonated hydroxyapatite (HAP), packed at high density (95 wt% in mature enamel), with only 1 wt% soft organic matrix and 4 wt% water[10,11]. As many other biominerals, it must be space-filling to withstand forces[12]. Enamel is a hierarchical nanocomposite material with an intricate organization[13,14], which is the key to its mechanical performance. The building block of enamel is the enamel rod—an array of aligned carbonated apatite crystals, which are thought to be oriented with their c-axes along the rod axis[15–17]. Notice that in this work we use the word *aligned* when referring to morphological alignment of elongated crystals, and *mis-oriented* or *co-oriented* when referring to the orientation of crystalline c-axes. Typical rod crystals in mature enamel are ~50 nm wide (26 nm × 63 nm according to Daculsi and Kerebel[18]) and more than 10 μm long, as previously reported[18–21]. The elongated crystals in each rod run parallel to one another[22–24]. Each rod is also associated with an interrod, which consists of crystals arranged at a ~60° angle to the rod axis[13]. Crystal elongation direction varies gradually from the rod to the interrod[25–27]. Each rod is wrapped in a sheath of organic matrix, whereas crystals within the rod abut one another, with discontinuous, organic meshwork in between[22]. Rods run from the dentin–enamel junction (DEJ) to the surface of enamel, with their trajectories undulating in the inner enamel layer, and thus creating Hunter–Schreger bands or a decussation pattern, previously imaged with electron microscopy, x-ray tomography, or x-ray fluorescence[28–31], which makes it 10 times more resistant than bone to crack growth[32–34].

Enamel in other mammals has different decussation patterns, for instance in mice all rods are parallel to one another in each layer, but layers alternate so rods are at 60° from one another[35]; in bovine enamel instead, crystallites within a rod are not simply parallel but twisting as fibers in a thread of wool[36].

Although the morphological organization of enamel is well understood[18,37], very little is known regarding how crystals are oriented *within* this organization, especially at the scale of tens or hundreds of microns, as high-resolution transmission electron microscopy (HR-TEM) and selected area electron diffraction (SAED) studies provide information from areas limited to the rod width[15,16]. Since the mechanical properties of the crystals differ in different crystallographic orientations[38,39], understanding this relationship provides important insights into the mechanical behavior of enamel. The methods used here include polarization-dependent imaging contrast (PIC) mapping[40–43], HR-SEM and HR-TEM. X-ray linear dichroism in apatite was discovered[44] and fully explored[35] recently. This effect enables the PIC mapping method used here. PIC mapping has been used extensively for carbonates[40,45–49], bone apatite[35,50], entire teeth[31], parrotfish enameloid[44], and mouse enamel[35]. In coral[48], sea urchin teeth[46,47], mollusk shell nacre[45], and prismatic calcite[49] the orientations measured by PIC mapping were confirmed in precisely the same regions with x-ray diffraction.

Here we analyze the structure of human enamel using all these methods, and the results reveal the enamel structure previously hidden. That is, the large angle spread of crystal c-axes within a rod and the small c-axis mis-orientations between adjacent crystals, which are needed to fully characterize and understand how this essential tissue in the human body works.

## Results and discussion

**30°, 60°, or 90° crystal mis-orientation within each rod.** The elongated crystals are indeed all parallel, and aligned with the long axis of the rod, but they are not co-oriented. In fact, the angle spread of their c-axes within a rod is typically 30° and occasionally up to 60° or even 90°, as shown in the PIC map of human enamel from a young adult molar, in the inner enamel region, presented in Fig. 1. For example, in three of the rods in Fig. 1a, the colors range from magenta (−60°) to blue (−30°) to cyan (0°). Even more strikingly, in Fig. 2 the colors in the rods cross-sectioned transversally range from red to black, and thus have an angle spread of 90°.

This means that, contrary to earlier reports, the long axis of each nanocrystal is not necessarily co-aligned with the crystalline c-axis, they can be as much as 90° apart. This is intriguing, since TEM studies found crystals in the enamel rods to be aligned with their long axes along the axis of the rod[15–17]. Apatite crystals elongate parallel to one another in enamel rods, as shown in both cryofractured enamel in Supplementary Fig. 1 and in etched enamel in Fig. 3 and Supplementary Fig. 2, as previously shown in many other experiments and schematics[14,27,51,52]. They gradually change in orientation from head to tail, as shown in Figs. 1 and 2, and previously assumed by many authors based on SEM imaging[25–27].

Imaging precisely the same area of enamel with PIC mapping and SEM, before and after etching, respectively, with the same magnification and orientation reveals that crystals elongating parallel to one another are majorly mis-oriented. The two images, presented in Fig. 3, are differently warped by the two microscopes, but they are recognizably from the same region of the tooth. Crystals in a rod that are differently oriented by almost 90°, that is, with green and magenta pixels in the PIC map of Fig. 3d, appear all approximately horizontal and aligned parallel to one another in the SEM image in Fig. 3c.

While it may be argued that the SEM image shows that the crystals are not exactly parallel to each other, the very small deviations in the crystal alignment are much smaller than the large (>30°) c-axis misorientations shown in the PIC map. Based on the higher magnification images in Fig. 3c, d, it is clear that the PIC maps indicate a large angular spread in c-axis orientations that does not correspond to a similar change in elongation direction in the SEM images.

Since normally HAP crystals grow along the c-axis, it has been assumed that enamel crystals also follow this growth pattern, yet PIC maps show clear evidence of a much greater angle spread compared to bright field TEM of enamel rods. The consensus model assumed co-orientation of long axis and c-axis, which is understandable, as enamel crystals have mostly been studied by electron diffraction in TEM from limited areas, where the $(0,0,\ell)$ reflections are only detectable if the crystals happen to have their c-axis perpendicular to the electron beam within ±1°[16]. Therefore, the only crystals whose orientation could be indexed were those that had their c-axes in plane in the TEM sample and along the rods, thus the analysis was strongly skewed towards this interpretation. In PIC mapping all orientations of c-axes are equally detectable, thus no skewing occurs. Careful re-analysis of previous TEM work[17,21] reveals that indeed the angle spreads observed here were seen before. Diverse orientations of crystals in a rod are consistent with the different electron densities of adjacent crystals in TEM observed by Selvig[21], which can be due to diffraction contrast across differently oriented crystals. Those data were not interpreted as diverse orientations as other

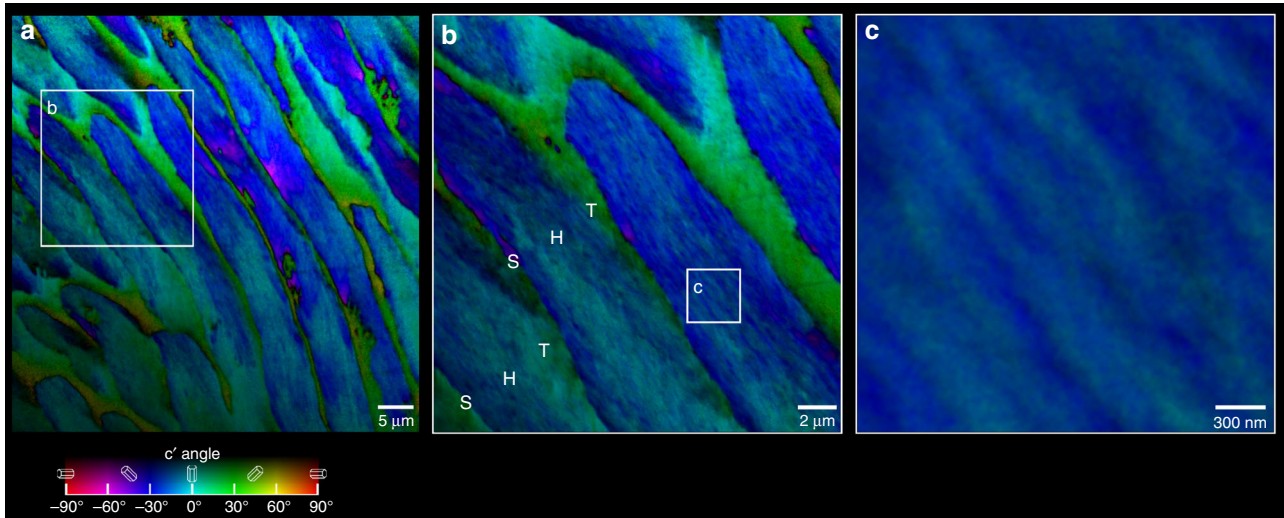

**Fig. 1** PIC maps revealing the hidden crystal orientation structure of inner enamel. **a** Low magnification map of polished cross-section of human enamel (see Supplementary Fig. 3 for the position of this area in the enamel polished cross-section). Notice the ~5 μm wide rods with a significant number of crystal c-axes oriented along the rod axis (blue). However, many other crystals are oriented ±30° off the rod axis (cyan and magenta). The c-axes of interrod crystals are highly co-oriented, as evident from the homogeneously green hue (+30° from the vertical in the lab and in this image) almost everywhere, with just a few orange pixels (+60°). **b** Zoomed-in PIC map acquired in the correspondingly labeled box in **a**, showing the fine details of the rod and interrod crystal orientation and arrangement. Notice in **b** that the transitions in crystallographic orientations between a rod head (H) and its interrod tail (T) is gradual, whereas the transition from the interrod to the next rod's head is abrupt and these are separated by an organic sheath (S). **c** Zoomed-in region in **b**, where individual crystals inside the rod are parallel to each other but their c-axes are not co-oriented, thus single or multiple co-oriented crystals stand out as different colors, e.g. blue surrounded by cyan or vice versa. Typical crystal width is ~50 nm, resolution and pixel size are both 22 nm in **b** and **c**, and 60 nm in **a**

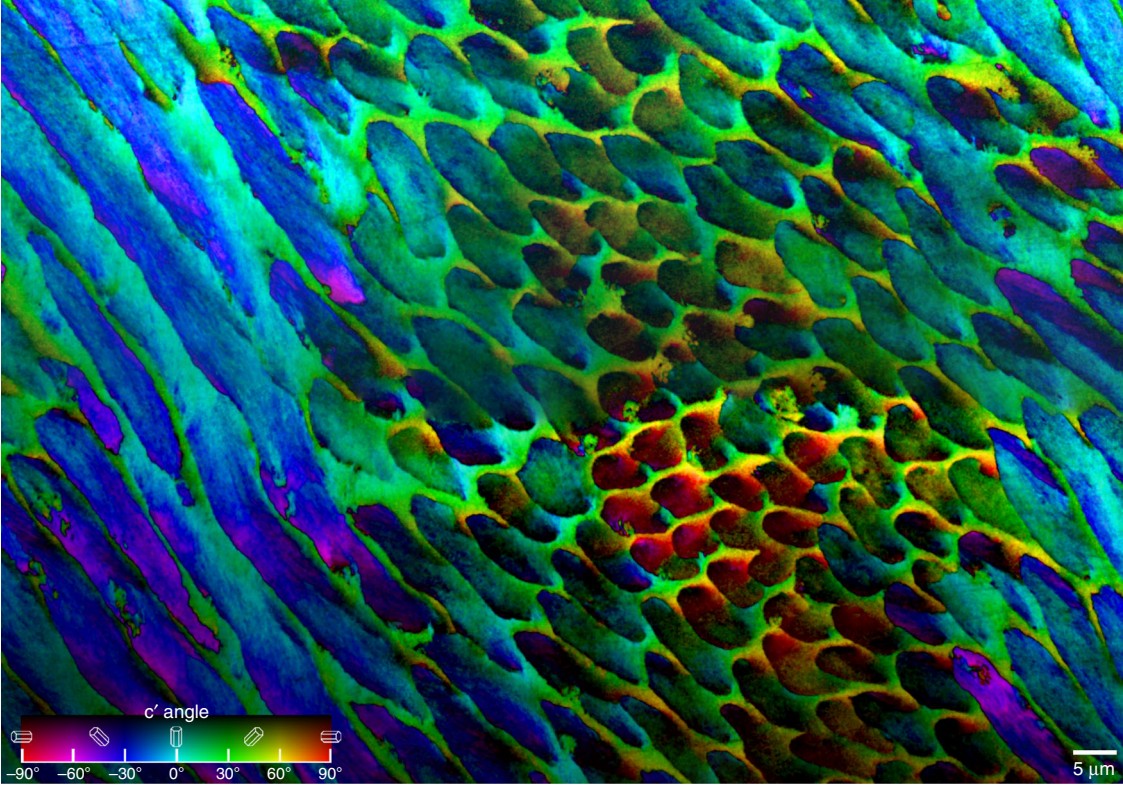

**Fig. 2** PIC mapping reveals the hidden crystal orientation structure in a large area of inner enamel. The map shows Hunter–Schreger bands, or decussation pattern, in inner enamel, with three groups of rods exposed on this polished surface: in longitudinal (left), transverse (right of center), and oblique (center, right) cross-sections. See Supplementary Fig. 3 for the exact position of this area in the tooth

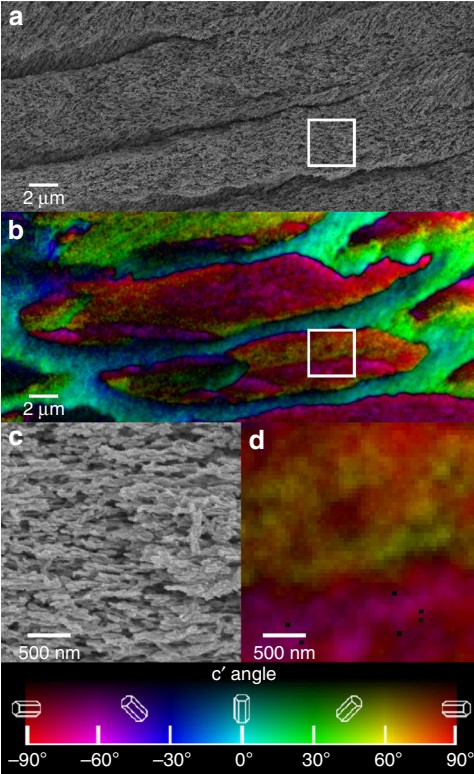

**Fig. 3** Comparison of SEM image and PIC map of the same region of human enamel. **a** The SEM image, acquired after etching, reveals two well-distinct rods, separated by interrod and deeper groves, and surrounded by other partial rods. **b** The PIC map shows that the same two well-distinct rods have multiple orientations within them, as all other rods imaged in this work. Since the bottom has more diverse orientations, we chose this rod to zoom-in further in (**c**) and (**d**), where the white boxes are located in (**a**) and (**b**) respectively. **c** Zoomed-in SEM image showing that all crystals are approximately horizontal and parallel to one another. **d** Zoomed-in PIC map showing that crystals from top to bottom of the box vary from red, to orange, to green in the top half, which is a 60° angle spread, and from red to magenta in the bottom half, which is a 30° angle spread. See Supplementary Figs. 4 and 5 for the exact position of this area in the tooth, and for the SEM image warping necessary to overlap precisely the bottom rod in the PIC map. This warping makes the top rod imprecisely correspond to the top rod in the PIC map

interpretations were possible, including different thicknesses, or other sectioning artifacts. In the present PIC maps all crystals are simply polished, hence their orientations are expected to remain as they were in pristine enamel. They did in the aforementioned carbonate biominerals[45–49]. The multiple orientations observed in each rod by PIC mapping are also fully consistent with the isolated, curved enamel crystals observed by Daculsi et al. extracted from enamel from human fetuses[19]. To corroborate the PIC mapping observations we did HR-TEM of thin sections of mature enamel and found that the $c$-axes of crystals in close proximity to one another within a $130\,\mathrm{nm} \times 130\,\mathrm{nm} \times 100\,\mathrm{nm}$ volume are mis-oriented by 23°, 27°, and somewhere between 18° and 90° (Fig. 4). These HR-TEM observations are in excellent agreement with and thus confirm the PIC mapping results.

The larger area of inner enamel shown in Fig. 2 was merged and blended from $3 \times 2$ partially overlapping PIC maps, acquired with 60 nm resolution. As in Fig. 1, all rods in Fig. 2 have a significant angle spread. The decussation pattern is a structural toughening mechanism, responsible for enamel's resistance to

crack growth[4,27,32]. In the decussation pattern observed in Fig. 2, the rods transversely or longitudinally sectioned show crystal orientations in the red-black or cyan-blue ranges of colors, as expected for rods that run approximately perpendicular to one another.

Within each rod in Figs. 1 and 2 one can frequently observe elongated nanocrystals oriented up to 30° from their immediately adjacent crystal, as shown in Fig. 1c, and evident within each and every rod in Fig. 2. We stress that the observed mis-orientations within rods are not due to the change in elongation direction of the rods in the decussation pattern: in that case crystal $c$-axes orientations would indeed change from rod to rod, but within each rod all crystallites should be co-oriented, but they are not. They never are, in any of the regions analyzed across the entire enamel layer, and across two different molars. Similarly, we are not focusing on the known and well-established gradual mis-alignment of crystals observed at the SEM from rod to interrod (also known as *from head to tail* within each *keyhole* unit)[25–27], but on crystal $c$-axes mis-orientations within each rod (head). The elongated nanocrystals within each rod are parallel to one another morphologically, as shown before by SEM and AFM[22–24], and by the present SEM data in Fig. 3, Supplementary Figs. 1 and 2. Their crystallographic orientations, however, vary dramatically, up to 90° across a rod (head). This means that the $c$-axis in some cases can be perpendicular to the elongation direction of the nanocrystals.

The $c$-axes angle that spread internal to each rod is never zero, and varies between 30° and 90°, as shown in all areas analyzed in this work, all of which are summarized in Supplementary Figs. 3 and 5.

**Mis-oriented adjacent crystals as a toughening mechanism**. We propose that mis-orientation of adjacent enamel nanocrystals provides a toughening mechanism. If all crystals are co-oriented a transverse crack can propagate across crystal interfaces, whereas if the crystals are mis-oriented a crack primarily propagates along the crystal interfaces, leading to material toughening via the crack deflection mechanism presented in Fig. 5.

A similar mechanism has been observed in metals after severe plastic deformation, where high-angle boundaries make metals strong, ductile, fatigue-resistant, and tough[53]. In metals, however, dislocations and sliding at grain boundaries have been invoked[53–55], which may or may not occur in enamel. The model for toughening we propose is simpler, as it only uses crystal orientations, which are directly and unambiguously observable.

Molecular dynamics (MD) simulations support the model proposed in Fig. 5a. The results are presented in Fig. 5b and Supplementary Movies 1, 2, 3, and fully described in the Supplementary Information, and Supplementary Figs. 6–10 and Supplementary Tables 1–3. The crystals were subjected to a pressure in the vertical direction of 1 GPa, which is comparable to that experienced by HAP crystals during mastication, assuming 1000 N chewing force and 1 mm² area of a tooth cusp. In all three cases the crystals sintered at 1 GPa pressure. The two sintered crystals are hereafter termed bi-crystals. Of course, there are trace amounts of water and proteins at some bi-crystal interfaces. These were omitted on purpose in our simulations, as it is well known that at such heterogeneous materials interfaces cracks are normally deflected[47,56,57]. The discovery here is that at interfaces of *the same material* cracks are deflected, provided the crystals are differently oriented. Koblischka-Veneva et al. did not observe any non-apatite material at grain boundaries in their electron backscatter diffraction (EBSD) of enamel, corroborating our interpretation that such materials are rare in enamel[58]. Thus, the simulations show crack propagation across co-oriented bi-crystal

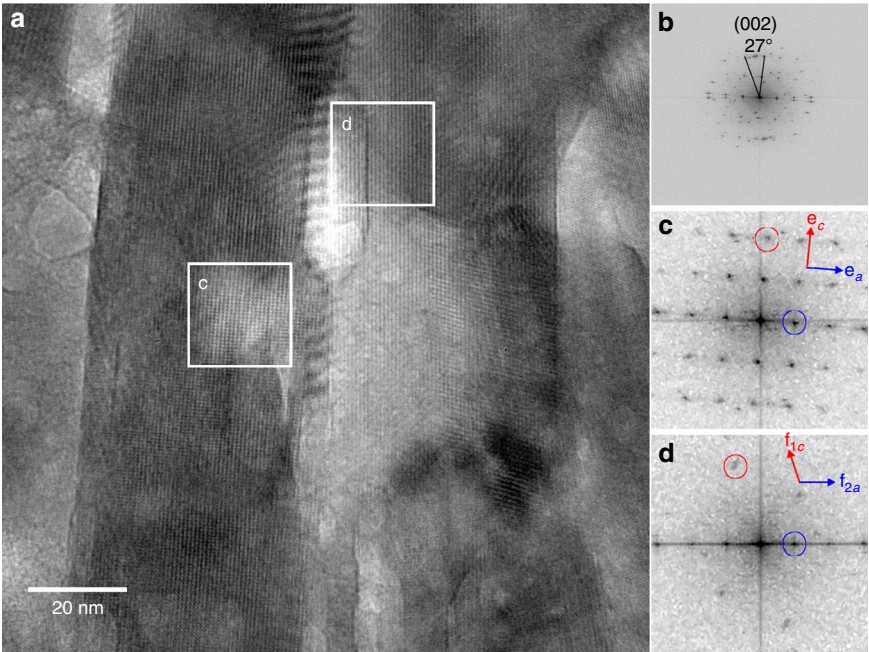

**Fig. 4** Crystal orientations of a thin section within a human outer enamel rod showing *c*-axis misorientation by 23°, 27°, and >18°. **a** HR-TEM micrograph taken from a 130 nm × 130 nm × 100 nm volume within an outer enamel rod, with crystals elongated in plane from top to bottom in (**a**) (termed *vertical* hereafter). **b** Fast Fourier transform (FFT) analysis of the entire image in (**a**), showing that two of the crystals within this entire volume have their *c*-axes mis-oriented by 27°. **e**, **f** FFT power spectra extracted from (**c**) and (**d**) in (**a**), which include crystals with their (100) planes almost parallel and vertical. In **e** and **f** red circles and arrows identify (002) spacings and *c*-axis directions, respectively; blue circles and blue arrows identify (100) spacings and directions of *a*-axes, respectively. The (**e**) FFT indicates the presence of a single crystal of carbonated apatite with its *c*-axis oriented 5° clockwise from the vertical and its *a*-axis at 90° from the *c*-axis as expected for apatite. The (**f**) FFT indicates the presence of two overlapping crystals, $f_1$ and $f_2$. The *a*-axis of crystal $f_1$ is horizontal (blue circle). No (001) lattice fringes were detected for $f_1$, thus its *c*-axis is out of the image plane in (**a**). The *c*-axis of crystal $f_2$ is oriented at 18° counterclockwise from the vertical (red circle). No (100) lattice fringes were detected for $f_2$, thus its *a*-axis is out of the image plane in (**a**). Crystal $f_2$ is oriented with its *c*-axis 18° counterclockwise from the vertical, thus the angle between the *c*-axes of crystals $f_1$ and $f_2$ is at least 18°, but it could be as large as 90°. The *c*-axes of crystals e and $f_2$ are 18° + 5° = 23° apart. Since enamel crystals are on average 26 nm × 63 nm in cross-section[18], the 30-nm wide crystals at the center of the image in (**a**) must be oriented nearly edge-on. Since this section is 100 nm thick, all three crystals identified in the FFTs are either in close proximity to or directly abutting one another. Thus, the *c*-axes of crystals in close proximity are 23°, 27° and somewhere between 18° and 90° apart. Supplementary Fig. 5 shows where the tooth sample was FIBed to extract this thin section

interfaces or deflection by mis-oriented bi-crystal interfaces, without either water or proteins.

The presence of apatite lattice defects and substitutions would affect the crystallite structure and therefore the mechanical response of crystals in MD simulations. These were omitted to keep the model as simple and informative as possible.

Most interestingly and unexpectedly, the simulations show that cracks behave differently depending on the mis-orientation angle. When crystals at a grain boundary are co-oriented ($\theta = 0°$) cracks propagate through the interface, when their *c*-axes are mis-oriented by 14° the cracks are deflected, but at 47° mis-orientation the crack again propagates through the interface. This result was reproduced in multiple simulations, using homogenous and inhomogeneous loading in the horizontal direction, and is therefore noteworthy, even though it was unexpected.

We quantified the critical energy release rate ($G_c$) (which is, despite this consensus name, a density in space, not a rate in time), also known as fracture energy, from the stress–strain curves in Supplementary Fig. 10A by integrating them to estimate the total external work necessary to break the entire bi-crystal system (strain $\varepsilon \approx 0.2$)[59]. We used this energy approach because the bi-crystals used in our simulations are not homogeneous systems. The critical energy release rate obtained is around 5.87 J/m$^2$ for co-oriented HAP bi-crystals ($\theta = 0°$), which corresponds to a fracture toughness $K_{IC} = 0.88$ MPa m$^{0.5}$ (using Young's

modulus $E \approx 133.3$ GPa), which is in good agreement with that of pristine HAP measured with nano-indentation, which is $K_{ICexper} = 0.65 \pm 0.14$ MPa m$^{0.5}$ [60]. The critical energy release rates for the bi-crystals with $\theta = 14.1°$ and $\theta = 47°$ mis-orientation increase to 8.6 and 7.4 J/m$^2$, respectively (see SI Methods for details). Thus, our MD simulation results show quantitatively that the energy necessary to fracture bi-crystals increases significantly, from ~6 to ~9 and back down to ~7 J/m$^2$, in the presence of mis-oriented interfaces.

**Small (1°–30°) mis-orientations better deflect cracks**. If indeed, as suggested by the MD simulations small mis-orientation angles are more effective at crack deflection than larger angles, is there a *sweet-spot* mis-orientation angle that maximizes energy release? Other mis-orientation angles could not be tested with MD simulations, due to periodic boundary condition constraints (see SI). Assuming that enamel's long evolutionary history may have selected for such a sweet spot, if one existed, the experiment to test its existence is simple: measuring the mis-orientation of *c*-axes in adjacent pixels in PIC maps. The histograms in Fig. 6 and Supplementary Fig. 11 demonstrate that most pixels are mis-oriented by 1° with respect to their neighboring pixels, and all of them are mis-oriented by <30° (Fig. 6). This may therefore be the sweet spot, that is, crystals 1–30° apart may maximize energy release and toughening.

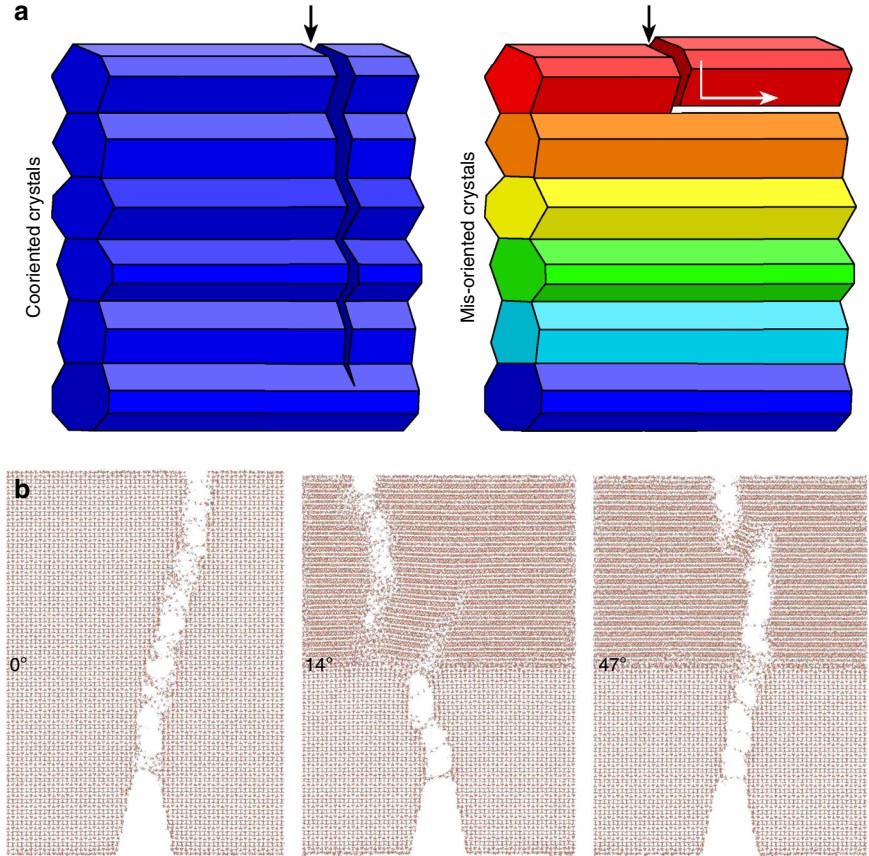

**Fig. 5** Crystal mis-orientation provides a toughening mechanism. **a** Schematic of the mechanisms: co-oriented crystals (blue) enable crack propagation across different crystals, precisely because they are co-oriented. When crystals are mis-oriented (colors), instead, cracks deflect at crystal interfaces, thus they cannot propagate or grow over long distances, and the material is tougher. **b** Molecular dynamics simulations of grain boundaries, where hydroxyapatite crystals are mis-oriented by 0°, 14°, or 47°. Notice that the crack starting from the bottom propagates straight through the 0° interface, is deflected at the 14°, and again not deflected at the 47° interface. See Supplementary Movies 1, 2, 3

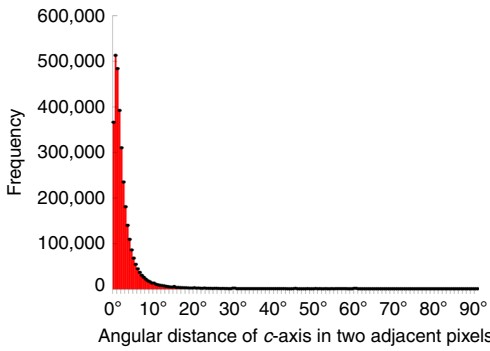

**Fig. 6** Histograms of angular distances of crystalline c-axes. The angular distances, in three-dimensional space, between the c-axes in each two adjacent 60-nm pixels, measured across all the pixels in Fig. 2. Almost all angular distances are below 30° and the peak is at 1°. Supplementary Fig. 11 shows additional histograms acquired every 2, 4, 8, 16, 32, 64, 128, or 256 pixels, demonstrating that orientations change gradually from pixel to pixel, thus from crystal to crystal within all enamel components. Small spikes around 30°, 40°, 45°, 50°, and 60° corresponds to rod–interrod interfaces without organic sheath

Remarkably, all crystals in human enamel rods are slightly mis-oriented with respect to their neighboring crystals, as shown by the histogram in Fig. 6. The few greatly mis-oriented adjacent pixels, e.g. 60°, occur at the rod–interrod boundaries where most mis-oriented crystals are separated by an organic sheath (non-polarization-dependent and therefore black in PIC maps) and only a few are not, generating the small spikes in the histograms of Fig. 6 and Supplementary Fig. 11. The model and simulations presented here predict that at greatly mis-oriented grain boundaries no crack deflection occurs, thus crack-deflecting organic sheaths[47,56,57] located at rod–interrod boundaries are necessary to toughen the material.

The finite element method (FEM) model in Supplementary Fig. 8 also confirms that small angles are better: tensile strain is lower at 14° than at 0° or 47° mis-orientations, and stress at the crack tip, under loading, is less concentrated at 14° than at 0° or 47° mis-orientations.

All observed orientation changes are gradual, as demonstrated by the data in Fig. 6: sampling adjacent pixels the mis-orientations are small (1–30°, Fig. 6), then they increase with sampling distance (Supplementary Fig. 11).

The zig-zag fracture in Supplementary Fig. 1D shows qualitatively the mechanical advantage of mis-orientation of crystals in enamel rods. Nanostructured materials[61,62] and crack deflection at mis-oriented interfaces limit crack propagation[63,64].

Crack deflection is a well-established toughening mechanisms[65], we therefore conclude that in enamel the observed mis-orientations play a key mechanical role: they increase the toughness of enamel at the nanoscale, which is fundamentally important to withstand the powerful masticatory forces, approaching 1000 N, repeated thousands of times per day[2].

Previous studies on the fracture behavior of enamel revealed other toughening mechanisms, such as microcracking, crack deflection, and branching at scales larger than the crystallites within a rod[66,67]. Thus all previous models have focused mainly on the role of protein–rod interface or rod alignment, always assuming that crystallites within rods were aligned and co-oriented, and thus could be treated as homogeneous[68,69]. Similarly, the effect of hydration or protein content previously reported[70] plays a role at the larger scale.

The fact that in human teeth fractures are not often observed across rods, but primarily at the micro-scale interrod–rod interface[67], demonstrates that the nano-scale toughening mechanism proposed here is effective.

The mis-orientations observed within all rods may result from imperfect oriented attachment of previously crystalline nanoparticles[71], which may be the nanoparticles observed after etching or freeze-etching by atomic force microscopy in human enamel[72], and were recently proposed by Robinson and Connell[73] to nucleate the larger crystals observed in mature human enamel. MD simulations show exactly that applying chewing pressure (1 GPa) to co-oriented or mis-oriented crystals makes the crystals fuse (sinter). Therefore, whether crystal fusion occurred during enamel formation as proposed by Robinson and Connell, or after the tooth erupted and started masticating, does not seem to be relevant to the function. What matters is that the crystals fused at some point, and are fused as the tooth masticates.

**Interrod crystals are co-oriented (0º–30°) for millimeters**. In contrast to rods, the interrod crystals are predominantly co-oriented throughout large areas of enamel, as shown by the nearly homogeneously green hue in Figs. 1 and 2 and irrespective of the axes of the rods. There are only a few pixels in which the interrod has a different orientation. One such exception is in the central region of Fig. 2, where the rods' axes are perpendicular to the image plane and the interrod crystal orientation is yellow (+60°) instead of green (+30°). Interrod enamel has long been speculated to be a continuous phase based on the alignment of its apatite fibers observed in SEM images[13,74–76]. All PIC maps in Figs. 1, 2, 3, 7, Supplementary Figs. 3, 4, 5, 12 show that not only are the interrod crystallites aligned, but their c-axes are highly co-oriented. This confirms that the term *continuous phase* used by previous authors[13] for the interrod was accurate. The near-co-orientation of all interrod is even more surprising when considering that each ameloblast cell deposits one rod–interrod complex with a head and a tail (H and T in Fig. 1b)[77]. All tails form a joined, co-oriented interrod continuum, thus many ameloblast cells must coordinate interrod deposition throughout vast areas. Analyzing the entirety of the enamel layer, which is 4 mm thick under the tooth cusp in Supplementary Fig. 3, we found that the same interrod orientation is conserved from the inner to the outer enamel, with just a few pixels of slightly different colors, from the aprismatic enamel (Fig. 7) at the surface, through the outer-, mid-, and inner-enamel, which are all shown in Supplementary Fig. 3.

In another tooth from a different young adult (Supplementary Figs. 5 and 12) we found two orientations of the interrod, each extending for 2/3 or 1/3 of the 1.7 mm enamel thickness under the cusp. Again, the aprismatic enamel at the surface is co-oriented with the nearby interrod crystals in Supplementary Fig. 5, as the interrod crystals are in Figs. 1, 2, and Supplementary Fig. 3.

**Aprismatic enamel crystals are randomly oriented**. Aprismatic enamel was expected to have c-axes perpendicular to the tooth surface. This is clearly not the case in two different teeth from different donors, in Fig. 7 and Supplementary Fig. 12, where the c-axes are +30° and −60°, respectively, that is, parallel and 30° from the surface of the tooth cusp. In order to rule out PIC mapping artifacts, in Supplementary Fig. 12 we present PIC maps of the second tooth mounted in two positions, rotated by 90°. After etching, in the same region of aprismatic enamel presented in Fig. 7, the crystals indeed appear perpendicular to the tooth surface, as previously observed[27] and as shown in Fig. 7, but their c-axes are almost parallel—not perpendicular—to the tooth surface. This is not the general orientation, in fact in the aprismatic enamel in Supplementary Fig. 12A, B we see the c-axis ~30° from the normal to the tooth surface, ~0° from the normal in Supplementary Fig. 12d–i, and ~66° from the normal in Fig. 7. The orientation, therefore, appears to be completely uncorrelated with the surface orientation.

The crystalline c-axes orientations observed in Fig. 7 are surprising because the crystals are all parallel to one another, and run perpendicular to the surface[27], thus they were expected to have their c-axes perpendicular as well. Their random orientation, however, is consistent with the data already seen in Figs. 2 and 3, where all rods have an internal angle spread ranging from 30° to 90°, thus crystals elongating perpendicular to their c-axis should no longer be unexpected. Interestingly, at the surface hardness and stiffness both exhibit their maximum values[78], and, at least in the case of Fig. 7, the maximum H and E occur across the c-axes not along them as previously assumed.

**Constraints for enamel formation future models**. The observation that the interrod is co-oriented over millimeter distances, whereas the rods elongate along various directions and, within those, exhibit various orientations, provides a strong constraint on any model for enamel formation. No current model of enamel formation can describe how a layer of connected ameloblast cells, each depositing 1 rod and 1 interrod, can achieve this geometry.

The crystal orientation of the interrod must be established once at the DEJ, and then be propagated through the growing interrod enamel layer either unchanged or changing rarely. At least three mineral growth processes could lead to the final co-oriented interrod: amorphous calcium phosphate precursors as observed in mouse enamel[79], with protein-guided particle attachment as shown by Fang et al. in vitro[80], or ion-by-ion precipitation of enamel crystals as described by Tomson et al. in vitro[81], or by formation of nanoribbons of amelogenin templating for the assembly of apatite crystals as proposed by Habelitz[82]. In any of these cases, interrod crystals' nucleation events must occur extremely unfrequently. Furthermore, organic molecules differ in rod and interrod during enamel mineral formation[83,84], perhaps contributing to the orientation differences observed here.

The mis-match of c-axis and elongation direction was observed in all crystals, within rods, interrod, and aprismatic enamel, sometimes by as much as 90°. The latter case does not mean that crystals grow along the a-axis or the b-axis direction. It appears that the crystal orientation in many cases is uncorrelated with the elongation direction, thus crystals can be oriented in any direction as they grow. This is consistent with two different formation mechanisms: (i) crystal growth via an amorphous calcium phosphate precursor phase[79], with the crystalline phase propagating through and at the expense of the amorphous phase, or (ii) crystal growth by imperfect oriented attachment of previously crystalline nanoparticles[71]. In both cases the organic matrix must exert significant control over the crystal growth to overcome thermodynamic constrains determining the crystal habit.

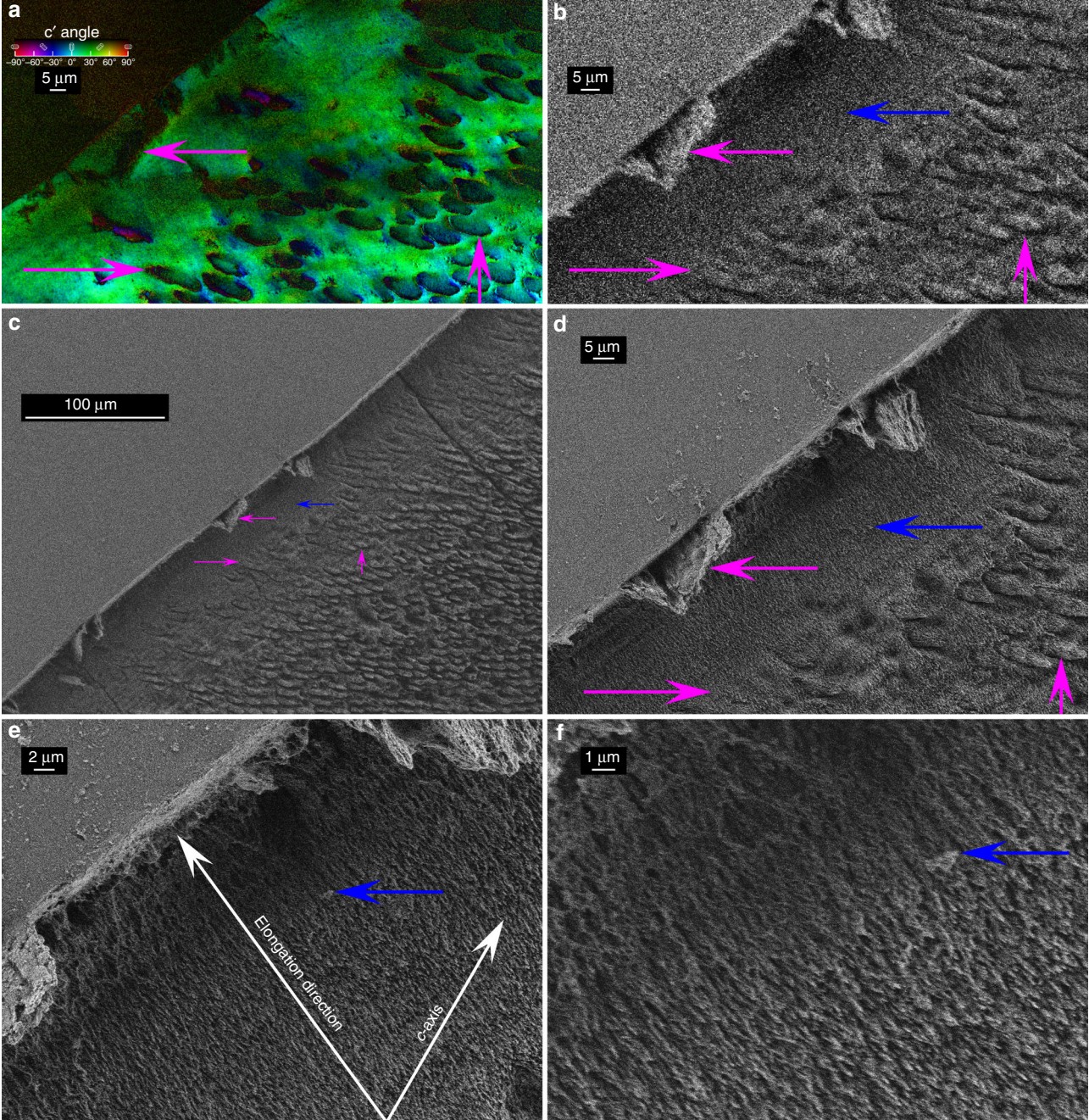

**Fig. 7** Aprismatic enamel at the tooth cusp surface. Notice that the aprismatic enamel is indistinguishable from the interrod, it just does not have any rods (previously termed prisms, whence the name of this aprismatic layer). See Supplementary Fig. 3 for the precise position of this region in the tooth. **a** PIC map of aprismatic enamel, showing that nearly all crystals are green, thus their c-axis is oriented at +30°. **b–f** SEM images of the same region after etching. **b** SEM image at precisely the same magnification as the PIC map in **a**, with magenta arrows indicating a hole in the tooth surface, infiltrated with epoxy, which resisted etching, and two rods. Arrows in **a** point to the same features before etching. **c** SEM image of the same region at lower magnification. The arrows were scaled down with the image, and indicate precisely the same features. **d–f** Increasingly magnified images of etched aprismatic enamel. The blue arrow in panels **b–f** indicate a feature visible in all SEM images and well resolved in **d–f**. Panels **e** and **f** clearly show that all crystals are aligned parallel to one another and perpendicular to the tooth surface. Panel **e** shows this as the *elongation direction*, which is −36° from the vertical. Their green color in **b** indicates a c-axis orientation of +30° from the vertical (also shown in **e**), thus the c-axes are 66° apart from the elongation direction, or 24° apart from the tooth surface. Thus, the crystalline c-axes are approximately parallel—not perpendicular—to the tooth surface. Supplementary Fig. 12 shows more PIC maps of the aprismatic layer in another tooth, confirming that crystalline c-axes are oriented randomly with respect to the tooth surface

**Comparison of mouse and human enamel**. Mouse incisal enamel was also analyzed with PIC mapping by Stifler et al. [35]. Comparing it to human enamel, we observe that in mouse enamel as well the c-axis orientation and the elongation directions do not match. In mouse inner enamel, however, each rod has a single orientation especially near the DEJ. Moving from the middle of the inner enamel towards outer mouse enamel, however, rods become gradually less homogeneous in

crystal *c*-axis orientations, and exhibit frequent gradual orientation changes.

All the data presented here provide a comprehensive and detailed understanding of the enamel structure, and provide a strong constraint on the three-dimensional geometry of human enamel formation. Crystals in the aprismatic and interrod enamel are highly co-oriented across the entire thickness of the enamel layer, whereas in the rods they are mis-oriented slightly (0–30°) with respect to their immediately neighboring crystals, and greatly (30–90°) across the rod in any orientation. The angle spread within a rod was never observed to be zero.

These data reveal the previously overlooked mis-orientation toughening mechanisms at play in human enamel, a most important biomineral for biting and mastication, and therefore for nutrition and survival of *Homo sapiens*. This structure, previously hidden, contributes to making enamel extraordinarily resilient, as it endures hundreds of mastication cycles per day, with hundreds of Newtons of biting force. This structure prevents catastrophic failure of enamel by deflecting cracks inside rods, and keeps it functional for our entire lifetime. Enamel and its crystal structure are well preserved in the fossil record[8,85], therefore an avenue for future discovery is to compare enamel structural evolution through time, and correlate it with known lifestyle and nutrition changes. More broadly, comparisons of enamel structures across mammals could explore structure–function correlations. Another avenue is to explore how wide-spread the mis-orientation toughening mechanisms is in biominerals and rocks, and how it can be applied to synthetic materials.

## Methods

**Samples**. Two healthy third molars extracted from two different patients for orthodontic purposes were collected at the Department of Oral Surgery, University of Pittsburgh School of Dental Medicine, and were used in this study. These are exempt from IRB approval. Only gender and age information was provided to the researchers. After extraction the teeth were cleaned, disinfected in 70% ethanol and stored at room temperature in air for a few months. They were then embedded in EpoFix (EMS, Hatfield, PA). The samples were cured at room temperature to avoid heat damage. They were then cut using a slow speed diamond saw (Isomet, Buehler, Lake Bluff, IL) along the plane of interest. The blade was water-cooled during sectioning to prevent potential heat damage. One tooth (Figs. 1, 2, 7, Supplementary Fig. 3) was cut along the buccolingual plane and through the tip of a cusp. Another tooth was cut along the occlusal plane, where one half of the tooth was used for FIB and the other half was cut again along the mesial–distal plane through the distobuccal cusp for analysis in PEEM (Fig. 3, Supplementary Figs. 4, 5, and 12). All samples for PEEM analysis were trimmed from the original 1" round embedding medium to ~15 mm × 15 mm squares, and 3 mm thickness. Three other teeth were cut along the buccolingual plane embedded, polished, and etch as described below.

**Polishing**. The samples were polished using 300 nm and then 50 nm alumina suspensions (Buehler, Lake Bluff, IL), saturated with HEPES buffer, pH 8.0 (Fisher Scientific, Waltham, MA) to prevent apatite dissolution. They were then rinsed in ethanol, air dried and coated with 40-nm Pt around the area of interest for PEEM experiments, and with 1-nm Pt in the area of interest while rotating and tilting the sample[86].

**Etching**. Three human third molars were embedded and polished as above, then coated with 1 nm Pt to emulate the PEEM samples, and then try to image the same areas previously analyzed in PEEM. They were then etched in 0.5 M EDTA in HEPES buffer, pH 8.0 (Fisher Scientific, Waltham, MA) for times varying from 5 s to 2 h. The 2-h time was the most effective for SEM imaging of parallel crystallites in each rod, showing crystallites well-separated from one another but with their positions not yet disrupted by excessive removal of material. Thus, this is the time used in the end for all samples. After etching the samples were rinsed twice in DD-$H_2O$ adjusted to pH 8, and once in pure ethanol. They were then dried with dry $CO_2$ and coated as described below.

The human molar shown in Fig. 3, Supplementary Figs. 4, 5, and 12 was gently polished using 50 nm alumina suspension (Masterprep, Buehler, Lake Bluff, IL) saturated with $CaCl_2$ to remove the Pt coating. Following common protocol for etching fossil teeth for SEM imaging[87–89], the tooth was then etched in 10 vol% HCl for 1 s, rinsed twice in DD-$H_2O$ adjusted to pH 8, once in pure ethanol, and dried with dry $CO_2$ and coated as described below. However, 1-s etching was not

sufficient for quality SEM imaging, so the tooth was gently polished again, and etched for 1 additional second as described above.

The human molar shown in Fig. 7 was also gently re-polished to remove the Pt coating, and etched in 10 vol% HCl for 2 s to reveal nanocrystal elongation direction.

**PIC mapping**. Tooth samples for PIC mapping were embedded into EpoFix, polished with alumina particles in down to 30 nm, and coated with 1 nm Pt in the area to be analyzed and 40 nm Pt around it to prevent charging[86,90] and enable photoelectrons escaping from the top 3 nm (at the Ca L-edge) of the sample to go through the 1-nm coating[91]. PIC mapping was done using the PEEM-3 microscope on beamline 11.0.1 at the Advanced Light Source (ALS) at Lawrence Berkeley National Laboratory, as recently described by Stifler et al.[35]. Thirty-eight images were acquired at 19 different polarizations between 0° and 90° in 5° increments. For each polarization angle, two images were acquired at anti-correlated energies, at the Ca L-edge ± 0.2 eV above and below 352.6 eV, which is the energy position of peak 1, and the best for enamel PIC mapping[35]. The images were spatially aligned in PEEMVision[92], then each image taken at the higher energy was digitally divided by that taken at the lower energy to maximize dichroic contrast at each polarization. The resulting stack of 19 images was used to create the PIC maps using the Polarization package in the GG macros[93].

In a PIC map the orientation of the crystalline *c*-axis in each pixel is measured and then displayed with different colors quantitatively corresponding to different *c*-axis angles. In all maps here the x-ray beam illuminates the sample from the right, at 30° grazing angle, and the linear polarization is rotated in the polarization plane, which is normal to the beam and intersects the sample surface and the image plane at a 60° angle. A vertical *c*-axis is in-plane in both the image and polarization planes. The in-polarization-plane component of the crystalline *c*-axis, termed *c'*-angle, is displayed as hue, with cyan being a vertical *c*-axis at 0°, and other angles as displayed in the color legend in Fig. 1a. The off-polarization-plane component of the *c*-axis, is displayed as brightness, shown as a gradient from bottom to top of the color legend in Fig. 1a. Full color means in-plane, and black 90° off-plane, that is, a black pixel indicates a *c*-axis pointing directly into the beam.

At the Ca L-edge, the maximum probing depth of PEEM and PIC mapping is 3 nm[91]. The photons penetrate 100 nm, but the photoelectrons only escape from the top-most 3 nm, of which 1 nm is Pt coating[86], and 2 nm is polished enamel. Therefore, the crystal *c*-axis orientation information is surface sensitive, and captures a single crystal per pixel. All PEEM data shown here were acquired with pixel size 57 nm × 57 nm × 3 nm, except for those in Fig. 1c, which had pixel size 22 nm × 22 nm × 3 nm.

Radiation damage during the PIC mapping experiments was minimal[94]. Charging phenomena are negligible[90].

**SEM**. SEM images in Supplementary Fig. 1 were collected using the Hitachi 5000 Field emission SEM at the Electron Microscopy Lab (EML) at UC-Berkeley, CA, USA. The enamel sample was notched, dipped in liquid nitrogen, then cryo-fractured at the notches, dipped into ethanol to prevent hydration of the surface from air moisture, as the sample thermalized to room temperature. The sample was then mounted on a stub with the cryo-fractured surface facing up, and coated with 20 nm Pt using a sputter coater (208HR, Cressington, UK), while rotating and tilting the sample to prevent charging of these highly topographic samples in the SEM.

The SEM images in Fig. 3 and Supplementary Fig. 2 were collected using a Zeiss LEO 1530 Field Emission SEM (Leo-1) at Nanoscale Imaging and Analysis Center (NIAC) at UW-Madison, WI, USA. All data in Supplementary Fig. 2 were acquired using the InLens detector, with 3 kV electrons, and 3.7 mm working distance. After etching and before SEM analysis the samples were coated with 20 nm Pt using a sputter coater (208HR, Cressington, UK), while rotating and tilting the sample to prevent charging. Samples were gently ashed (Mobile Cubic Asher (MCA), IBSS Group, Phoenix, AZ) for 10 min at 85 W to prevent carbon contamination. Identification of precisely the same area as in the PEEM data was not possible, due to the low contrast in polished sample at the SEM. All SEM data, however, were acquired in regions no more than 1 mm from those imaged in PEEM.

**Focused ion beam (FIB)**. FIB was performed on one sample using 1540XB CrossBeam® Zeiss Auriga Focused Ion Beam FE SEM containing a Ga liquid metal ion source. The sample was the same as in Fig. 3, Supplementary Figs. 4, 5, and 12, where an area near the cusp surface was selected, and a 20 µm long, 100 nm thick section was cut by FIB. This region was cut from the region where the rods appeared in cross-section as round as possible, thus the probability that they ran perpendicular to the polished surface was maximized, which enabled the FIB section to contain only in-plane rods. The precise position of the FIBed section is indicated in Supplementary Fig. 5, where it is labeled *FIB for 4*. The TEM image in Fig. 4 and many others confirm that indeed most crystals had their long axes in plane, and running from top to bottom in Fig. 4a.

**TEM analysis**. HR-TEM was conducted using a JEOL 2020 TEM (JEOL, Peabody, MA) equipped with a Schottky field emission gun, and a GIF TRIDIEM post-column energy filter (Gatan, Warrendale, PA) operating at 200 kV.

**TEM data processing**. Fast Fourier transforms (FFT) of either the whole image or selected portions thereof and the analysis of the power spectra including indexing were carried out using the ImageJ package (ImageJ, Bethesda, https://imagej.nih.gov/ij/).

**Molecular dynamics simulations**. To understand the behavior of cracks across interfaces of mis-oriented HAP crystals, MD simulations were performed via a LAMMPS package[95]. We utilized a previously developed interatomic potential for HAP[96] to describe the mechanical properties. The potentials utilize Coulombic charge interaction and Buckingham potential for non-bonded interactions. Morse-type potential is used for O–H and P–O bond and three-body potential is used for the angle O–P–O. The calculation of long-range Coulomb interaction is critical to describing the dynamics because HAP is an ionic crystal. To account for Coulombic interactions, we made use of particle–particle particle–mesh (PPPM) in all simulations. The lattice parameters and elastic constants obtained in the current study show good agreement with those from both experiments and theoretical calculation, including density functional theory (DFT) as seen in Supplementary Tables 1 and 2. The potential can describe mechanical properties of pristine and amorphous HAP[97,98], which is crucial for the interface between two crystals.

We generated three models, each containing two crystals, as shown in Supplementary Fig. 6. HAP in the current simulations with the Morse and Buckingham type potentials is very sophisticated. For example, the bond with Morse-type potential can easily lose the bond length during the structural manipulation and energy minimizations, which results in simulation crashes. Thus, after we abruptly changed the structures we changed the bond type to harmonic, and relaxed the structures for 1000 steps before further simulations with the original potentials. The process only adjusts all bonds to their equilibrium lengths, without affecting the simulation results because we relaxed again with the original potentials. We found that this process of temporarily changing the bond type to harmonic, and structure relaxation is crucial for the stability of simulations.

The mis-orientation angles in the top crystal were chosen to have a periodicity in the $z$ directions based on the equations, $l_x \sin\theta = a_z n$, where $l_x$ is the length of HAP in the $x$ direction; $a_z$ is the lattice length in the $z$ direction (6.86 Å); and $n$ is an integer number. For $n = 1$, 2, and 3, we obtained the mis-orientation angles to sustain the periodicity in the $z$ direction as 14.1°, 29.2°, and 47.0°. However, we chose only 14.1° and 47.0° as the low angle and high angle mis-oriented crystals, respectively, because the periodicity in the $x$ direction was broken with 29.2°. One may get different angles with different system sizes. However, the angles would be also restricted to discretized values if the system size is consistent. Thus, we decided to perform MD simulations with these three mis-orientation angles ($\theta = 0°$, 14.1°, and 47.0°), and to compare the crack behavior at these angles. We utilized a previously developed code for polycrystalline structure of graphene to build initial bi-crystal geometries[59]. Due to the potentials and long-range interaction, building bi-crystal models with a sharp crack for tensile tests requires well-designed steps.

First, we prepared a unit cell after structural relaxation at 300 K and 1 bar with NPT ensemble. We confirmed that the unit cell structure was fully relaxed based on its root-mean-square-deviation (RMSD). Utilizing the unit cell, we built initial configurations for bi-crystals as shown in Supplementary Fig. 6A. Then, the structures were relaxed at 1 bar and 300 K for 50 ps with 1 fs time steps. We applied four stages with different pressures in the $y$ direction to build well-sintered interfaces. For stage 1, we increased the pressure in the $y$ direction from 1 bar to 1 GPa for 10 ps; for stage 2, we maintained the 1 GPa pressure for 10 ps; for stage 3, we decreased the pressure to 1 bar from 1 GPa for 10 ps; for the final stage 4, the structures were further relaxed for 10 ps at 1 bar. We repeated this cycle five times and thus obtained well-sintered bi-crystals of HAP.

Second, we inserted vacuum between the nearest image cells, that is, the cells are copied from the simulation box to satisfy the periodic boundary conditions for calculations by stretching the box in the $y$ direction more than 20 Å as indicated with blue boxes in Supplementary Fig. 6C. In order to insert a sharp crack tip in the bottom crystal, we need to have enough space to ignore the interaction between image cells in the $y$ direction. During the box change, we fixed only one atom of molecules in the $y$ direction. For example, only P atoms were fixed while O atom did not have any restriction. All Ca atoms were fixed in the $y$ direction. This fixation is important to safely separate the interface between image cells in the $y$ direction. We successfully conserved the centered interface, while the interface between the image cells was clearly separated. After the vacuum was inserted, the structures were further relaxed at 300 K with NVT ensemble.

Third, a triangular crack was introduced (~5 nm) at the bottom of the bottom crystal ($\theta = 0°$). The atoms and molecules were firstly deleted from the sharp triangle as shown in Supplementary Fig. 6C. The system after this crack insertion was not charge neutral. Therefore, we adjusted the numbers of molecules based on their charges. We set a 6 Å skin around the crack, and removed more atoms when they needed to be deleted for charge neutrality. Then, the structure was further relaxed at 300 K with NVT ensemble.

Fourth, after preparing sintered bi-crystals with sharp cracks, we applied tensile loading along the $z$ direction to observe crack propagation. Before applying tensile deformation by stretching the box at a rate of 0.05 Å/ps, the atoms at the boundaries were fixed in the $y$ and $x$ directions. Again, only one atom of each molecule was fixed to obtain stable simulations. If all atoms were fixed, the system

easily crashed due to the unphysical rotation of OH or $PO_4$ molecules. The stress–strain curves were recorded. The 14.1° mis-oriented bi-crystal shows a clear crack deflection and thus toughening (see Supplementary Movies 1–3). Since the system size for these MD simulations was very small (~20–30 nm), one may question about the homogenous loading condition. Therefore, we designed inhomogeneous loading conditions based on the FEM results for a system size close to the observed scale in real enamel.

FEM was employed to investigate the strain distribution near the crack tip, which was used as a boundary condition in the MD simulation. For FEM simulations, we built a 2D thin plate model with 10 layers of HAP, each with a height of 100 nm, and different orientations. It is noted that the layer height of the model is not crucial as we have tested heights from 50 to 150 nm using the FEM and basically the strain field within tens of nanometers of the crack tip is not significantly affected. The overall dimension of the FEM model is $8\,\mu m \times 1\,\mu m$, as illustrated in Supplementary Fig. 7A. Layer 0 is the first layer at the bottom of the model with a triangular crack at the bottom edge with the same shape as for MD simulations. The crack is 90 nm in height, leaving 10 nm space above it from the boundary with Layer 1 (Supplementary Fig. 7B). To make sure that the deformation in front of the crack can be accurately captured, we increased the element density from the crack tip to the grain boundary (Supplementary Fig. 7B). Convergence test was performed to ensure the proper mesh sizes. Considering that HAP has anisotropic mechanical behavior and the crystal orientation in each layer can be different from its neighboring layer, we assigned individual material property to each layer of the model according to Vannucci:[99]

$$C_{l=i} = [R(\theta_i)][C_{l=0}][R(\theta_i)]^{\mathrm{T}}, \tag{1}$$

where $[R]$ is the rotation tensor that is defined by the rotation angle of the layer $i$ ($i = 0$–9) along the $y$-axis from the first layer, which is given by

$$\theta_i = i \times \Delta\theta, \tag{2}$$

where $\Delta\theta$ (with a value of 0°, 14.1° and 47°) is the mis-orientation between the two neighboring layers. $[C_{l=0}]$ is the elastic tensor for the anisotropic elasticity of the HAP, which each of the elastic terms calculated from MD simulation as given in Supplementary Table 2.

$$C_{l=0} = \begin{bmatrix} C_{11} & C_{12} & C_{13} & 0 & 0 & 0 \\ & C_{11} & C_{13} & 0 & 0 & 0 \\ & & C_{33} & 0 & 0 & 0 \\ & & & C_{44} & 0 & 0 \\ & \mathrm{Sym} & & & C_{44} & 0 \\ & & & & & C_{11}-C_{12} \end{bmatrix} \tag{3}$$

We applied a simple boundary condition by uniformly stretching the left and right edges of the model by 0.5 μm of deformation using a static loading. We used the implicit solver of the open-source FEM program Calculix[100] to solve for the deformation field. The FEM simulation results are summarized in Supplementary Fig. 8. It is shown that the stress distribution in front of the crack tip is significantly affected by the material orientation and the grain boundaries within HAP can cause the stress to be totally discontinuous (Supplementary Fig. 8C) rather than the continuous stress field with a typical butterfly shape (Supplementary Fig. 8A). Based on the FEM result of the deformation field, we measured the displacement field around the crack tip by defining a region with width $w = 20$ nm and height 30 nm (starting 5 nm below the crack tip) before the deformation, and measured the relative elongation of the left and right edges of the region after the deformation ($\Delta w(y)$) as a function of the $y$ coordinate. This result, normalized to the original width ($\Delta w/w$), is the tensile strain ($\varepsilon$) that is summarized in Supplementary Fig. 8D for different $\Delta\theta$ values. These discrete measurements were fitted with an exponential function (Supplementary Fig. 9 and Supplementary Table 3), which was then used to apply the inhomogeneous tensile load to MD simulations.

The stress–strain curves from both homogeneous and inhomogeneous tensile loads are shown in Supplementary Fig. 10. For inhomogeneous load, we had to add vacuum in the $z$ direction based on the second step we described for the interface separation. Both homogeneous and inhomogeneous loads show similar trends. The model with 14.1° mis-oriented bi-crystal shows crack deflection, and thus toughened behavior of stress–strain curve. We tested strain rate effects with 0.5× and 2× and confirmed that there was no significant difference.

In MD simulations as well the results of applying homogeneous and inhomogeneous tensile loads are the same, thus in Fig. 5 and in Supplementary Movies 1–3 we present the homogeneous load results.

**Angular distances**. The RGB values of every pixel in the image in Fig. 2 were recorded to a file using the *Save XY Coordinates* function in ImageJ (ImageJ, Bethesda, MD). A program we developed in Xcode took the RGB value of each pixel and converted it to HSB. The hue (H) was related to the in-plane angle and the brightness (B) to the out-of-plane angle, so each pixel was represented as a unit vector with its HB angles. The angular distance between each pixel and its neighbors to the right and below was calculated using the dot product in

component form set equal to the cosine of the angle between the two unit-vectors. All angular distances were saved and plotted as a histogram with a bin size of 0.5° to produce the histograms in Fig. 6 and Supplementary Fig. 11. Histograms were plotted in Kaleidagraph® 4.5 for Mac for every pixel, every other pixel, every 4th, and so forth doubling the distance in each histogram up to 256. After importing all histograms into Adobe Photoshop® CC 2017, the 10,000 frequency was chosen arbitrarily, then the angular distance at this frequency was measured in each histogram, and plotted at the bottom of Supplementary Fig. 11, again using Kaleidagraph®, and importing into Photoshop®.

## Data availability

Source Data Files are provided as stacks of P3B images on https://figshare.com/projects/The_Hidden_Structure_of_Human_Enamel_Nature_Communications_2019/67034.

Each data set is provided as an Igor Pro experiment with file extension.pxp and as a stack of 38 unratioed, unaligned images with file extension .P3B. All .pxp files for a figure are zipped together in a folder, and each stack is zipped separately. They are grouped and named as follows:

Fig. 1: H247 and H250. Acquired in September 2017.
Fig. 2: Source data are provided as Source Data files H276, H279, H283, H284, H286, H288. Acquired in September 2017.
Fig. 7: Source data are provided as Source Data files H108, H111, H113, H116, H118, H119. Acquired in September 2017.
Supplementary Figure 12: Source data are provided as Source Data files H03, H04, H07, H08, H10, H11, H12, H13, H14, H15, H153, H154, H155, H156, H157, H158. Acquired in Jan. 2018. The bottom 6 panels are from Source Data files H335, H338, H341, H342, H345, H346. Acquired in April 2018.

## Code availability

The Igor Pro macros, called GG Macros, used to produce PIC maps are available free of charge on https://home.physics.wisc.edu/gilbert/software/. The code to measure the angular distances of c-axes in Fig. 6 and Supplementary Fig. 11 is available on https://home.physics.wisc.edu/gilbert/software/ and on https://figshare.com/projects/The_Hidden_Structure_of_Human_Enamel_Nature_Communications_2019/67034.

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

## Acknowledgements

The data reported in the paper are stored on Box and Dropbox and are available upon request. We thank Adrian B. Mann for discussions, P.U.P.A.G. acknowledges support from the U.S. Department of Energy, Office of Science, Office of Basic Energy Sciences, Chemical Sciences, Geosciences, and Biosciences Division, under Award DE-FG02-07ER15899, and NSF grant DMR-1603192. G.S.J., Z.Q., and M.J.B. acknowledge support from ONR (N000141612333) and AFOSR (FATE MURI FA9550-15-1-0514), as well as NIH U01EB014976 and U01EB016422. PEEM (BL 11.0.1.1) experiments were done at the Advanced Light Source, which is supported by the Director, Office of Science, Office of Basic Energy Sciences, of the U.S. Department of Energy under Contract No. DE-AC02-05CH11231.

## Author contributions

P.U.P.A.G. conceived the experiments and assembled the group, E.B. provided the human teeth, C.A.S. prepared them for the experiments, P.U.P.A.G., E.B., C.A.S., C.-Y.S. collected data, P.U.P.A.G. processed data, prepared all figures, and proposed the toughening mechanism model in Fig. 4a, contacted M.J.B. who agreed to do theoretical modeling to test it, M.J.B. and G.S.J. did molecular dynamics simulations, M.J.B. and Z.Q. did finite element simulations. E.B. collected and analyzed HR-TEM data and prepared Fig. 4. CAS collected SEM images in Figs. 3, 7, Supplementary Figs. 2 and 4, PG collected those in Supplementary Fig. 1. P.U.P.A.G. and E.B. co-wrote the manuscript, all other authors edited it.

## Additional information

**Competing interests:** The authors declare no competing interests.

