## [Peer Review File · Nature Communications]

Reviewers' comments:

Reviewer #1 (Remarks to the Author):

The authors applied innovative methods to take a new look at the microstructure of enamel crystallite organization in human teeth. The focus of the manuscript is on the application of X-ray circular dichroism analyses to discern between enamel crystallite shape arrangement, specifically the crystallite long axis, and the crystallographic c-axis of the carbonated hydroxyapatite crystallites in situ.

The paper questions the current status quo understanding in the field of enamel research that the long axis of enamel crystallites is co-aligned and co-oriented with the crystallographic c-axis. This is an interesting question indeed, however, the presented data do not support the claimed distinction between crystallite long axis and crystallographic c-axis. Consequently, the interpretations and molecular dynamics modeling, do not support the conclusion that the mis-orientation of crystallographic c-axes between the crystallites within a given enamel prism provides a mechanism to minimize crack propagation in the tooth crown.

The application of PIC mapping is novel for the analysis of human tooth enamel but has been explored and published in a recent publication in JACS (Stifel et al. 2018) using mouse enamel. The current understanding in the field is that murine enamel organization differs from human enamel in its arrangement of enamel rods, for example the decussation angle of rods is bigger in mouse than in human enamel. However, the fundamental process of tooth enamel formation has been shown to be conserved among mammals, including mineral phase, organic matrix components, crystallite size, shape, and organization into rods.

While the authors raise a very interesting question indeed, the presented evidence does not support the interpretation that the long axis of enamel crystallites is different from the crystallographic c-axis. This is mostly due to the lack of data such as SEM images at the same scale and location as PIC maps that would allow for the direct comparison between shape of crystals (alignment of crystal shape) and crystallographic c-axis seen in PIC mapping. Importantly, that the structural organization of enamel crystallites into undulating prism and inter-prismatic enamel it is known and very well described for many species and illustrated in many publications. Based on this body of literature one would expect the crystallite axis and alignment between crystallites to change continuously, that is to go in and out of plane of sectioning, a pattern known as Hunter-Schreger-bands, as well as to change their orientation within prisms.

A body of literature describes the hierarchical arrangement of tooth enamel and enamel crystal dimensions and phase.

For example: crystallite dimensions with thickness in the 20nm range, width being below 100nm (Selvig 1972)
(Kerebel et al. 1979)
(Daculsi et al. 1978, Daculsi et al. 1984)

Fracture behavior of enamel under consideration of crystallite alignment and enamel prism organization (Scheider et al. 2015, Yilmaz et al. 2015, Yilmaz et al. 2015).

Synchrotron analyses tracking enamel prisms from the DEJ to the enamel surface to show the cork-screw paths of enamel rods. The implication is that at any plane of section the enamel prisms, especially in the zone of inner enamel, and crystallites within the prism will be in plane for only a short path length and undulate in and out of plane of section (Tafforeau et al. 2012).

SEM data illustrating bovine enamel microstructure, including decussation pattern of prism and

crystallites within prisms not being in a simple parallel arrangement but organized similar to the twisting fibers in a thread of wool (Wang et al. 2012).

SEM images showing human enamel microstructure, including decussation pattern and Hunter Schreger Bands, that is the undulating of enamel prisms (Risnes 1998).

Methods description and application of linear dichroism to apatite, in particular mouse enamel prisms and crystal arrangement in mouse enamel (Stiffler et al. 2018)

Relevance and effects of hydration status of enamel and protein content on the mechanical properties (Baldassarri et al 2008).

The following further evidence would be required to strengthen the conclusions.

- SEM images at the same scale and ideally the same location as the PIC mapping figures to show the arrangement of crystallites and that the crystallite long axis does not coincide with crystal c-axis orientation
- TEM sections or FIB-lift outs that are either thinner than 100 nm to not include several crystallites superimposed, or are at 100 nm thickness and include single crystallites
- The thickness of enamel crystallites is in the 20 nm range (Kerebel et al. 1979), and the TEM images in the present manuscript show superimposed apatite crystals. The diffraction patterns are, therefore not of single crystallites. In addition, the diffraction pattern C does not show the clear orientation as indicated by the arrow in Figure S3.
- Methods should provide an adequate protocol for sample preparation, although it is known that dehydration and loss of organic material affects the mechanical properties of tooth enamel (Baldassarri et al 2008).
- The methods need to include which teeth were actually used, whether they were extracted for orthodontic reasons and with IRB approval, whether the enamel had any defects, whether they had been stored and how, were fixed and/or dehydrated, what the exact position and plane of sectioning was.
- The authors state, based on their molecular dynamics modelling, that mis-orientation of adjacent crystallites induces crack deflection. Yet, the model ignores the fact the interface between crystallites is not perfect and contains protein and water.
- The modelling approach and data interpretation needs not acknowledge that the presence of lattice defects and substitutions can affect the crystallite structure.

Several publications provide in depth explorations of the mechanical properties of tooth enamel under consideration of prisms and crystallite orientation and presented experimental data (Scheider et al. 2015, Yilmaz et al. 2015, Yilmaz et al. 2015). These studies have more immediate and biological relevance than the modelling data provided in the presented manuscript, that ignore presence of water and organic molecules between crystallites.

Bibliography;

- Baldassarri M, Margolis HC, Beniash E. 2008. Compositional determinants of mechanical properties of enamel. *J Dent Res.* Jul;87(7):645-9
- Daculsi G, Kerebel B. 1978. High-resolution electron microscope study of human enamel crystallites: size, shape, and growth. *J Ultrastruct Res.* 65(2):163-72.
- Daculsi G, Menanteau J, LM; K, D; M. 1984. Length and shape of enamel crystals. *Calcif Tissue Int.* 36(5):550-5.
- Kerebel B, Daculsi G, Kerebel LM. 1979. Ultrastructural studies of enamel crystallites. *J Dent Res.* 58(Spec Issue B):844-51.

Risnes S. 1998. Growth tracks in dental enamel. *J Hum Evol.* 35(4-5):331-50.

Scheider I, Xiao T, Yilmaz E, Schneider GA, Huber N, Bargmann S. 2015. Damage modeling of small-scale experiments on dental enamel with hierarchical microstructure. *Acta Biomater.* 15:244-53.

Selvig KA. 1972. The crystal structure of hydroxyapatite in dental enamel as seen with the electron microscope. *J Ultrastruct Res.* 41(3):369-75.

Stifler CA, Wittig NK, Sassi M, Sun CY, Marcus MA, Birkedal H, Beniash E, Rosso KM, Gilbert P. 2018. X-ray Linear Dichroism in Apatite. *J Am Chem Soc.* 140(37):11698-704.

Tafforeau P, Zermeno JP, Smith TM. 2012. Tracking cellular-level enamel growth and structure in 4D with synchrotron imaging. *J Hum Evol.* 62(3):424-8.

Wang C, Li Y, Wang X, Zhang L, Tiantang, Fu B. 2012. The enamel microstructures of bovine mandibular incisors. *Anat Rec (Hoboken).* 295(10):1698-706.

Yilmaz ED, Jelitto H, Schneider GA. 2015. Uniaxial compressive behavior of micro-pillars of dental enamel characterized in multiple directions. *Acta Biomater.* 16:187-95.

Yilmaz ED, Schneider GA, Swain MV. 2015. Influence of structural hierarchy on the fracture behaviour of tooth enamel. *Philos Trans A Math Phys Eng Sci.* 373(2038).

Reviewer #2 (Remarks to the Author):

This paper reports on observations of the microstructure of human dental enamel using a novel analysis technique called PIC-PEEM. This technique allows the detection of crystal orientation in association with EM imaging over larger areas. The authors state that they discovered new structural elements related to the alignment of fibrous apatite crystals in enamel: A) continuous alignment of fibrous apatite crystal in the interrod enamel; B) mis-orientation and gradual change of apatite orientation within an enamel rod; C) aprismatic enamel at the occlusal surface is not comprised of apatite fibers perpendicular to the surface. In addition MD simulations were performed to demonstrate the role of mis-alignment of crystals in crack propagation.

Some of the information the authors provided is novel. In particular observation C runs contrary to what is mostly stated in the literature. The current view is that the outer aprismatic enamel has apatite crystals oriented in parallel with the c-axis perpendicular to the occlusal surface. In the PIC map provided in Fig 4, the aprismatic enamel has the same orientation as the interrod enamel which is intriguing and plausible. This is a significant novel observation. To ensure any artifact it would be beneficial to obtain a TEM or SEM image of the surface zone and demonstrate the alignment of the enamel crystallites with regards to the occlusal surface.

Observation A, refers to the interrod enamel. Interrod enamel has long been speculated to be a continuous phase and many SEM images have shown such an alignment, see work by Warshawsky in the 70s and more recent work by Shane White, the Ariola group and Schneider group on fracture mechanics (see refs in the manuscript). The PIC map of Figure 4 adds additional information and more clearly demonstrates the alignment of apatite fibers through the thickness of the enamel and confirms the term continuous phase used by previous authors.

Observation B, was surprisingly characterized as a novel observation. While certainly in simplified models that describe the mechanical behavior of enamel a parallel alignment within rods has been used to ease simulation calculations, it is widely reported that apatite fibers in enamel rods are not aligned in parallel across an enamel rod. In fact most current oral/dental histology textbooks still use the illustration by Meckel from 1965 (*Arch Oral Biol.*) to illustrate the change in orientation from head to tail within a keyhole-like structure (see Ten Cate textbook referenced in manuscript), see also Boyde 1967. It is surprising that this existing model was overlooked by the authors. While Meckel did not believe in the presence of interrod enamel, which is incorrect in his model, he correctly described the lateral flare of apatite crystals across enamel rods which has been shown in many studies involving SEM analysis. A benefit of this orientation to crack deflection is somewhat obvious, but has,

to this reviewer's knowledge, never been considered previously. However, clinically and in vitro observations clearly suggests that cracks follow the long axis of the enamel rod and run through the interrod-rod interface in enamel. An analysis of the effect of the about 30 degree oblique angle of interrod enamel to crack propagation appears more relevant to our understanding of cracking in enamel.

Also, of concern in the MD simulations of cracks within a rod is the lack of evidence of fine enough crack radii that actually are able to interact between the enamel crystallites. In this regards the statements that enamel crystallites measure $\sim 100\text{nm}$ is not correct. It is well documented that crystal measure about $\sim 50\text{nm}$ in cross sections ($40\text{-}60\text{nm}$). Therefore the assignment in Fig. 3C of changing crystal orientations at domains of about 100nm is not due to single crystal mis-alignments.

Details:

1. Information about the thickness of the two specimens used for PIC-PEEM analysis was not provided. In this regards it would be of keen interests to provide information about the penetration depth of the X-ray analysis and its impact on the data collected was not provided, but are critical to the interpretation of the findings. Also some description of the method used would be useful to the reader.

2. Microns should be micrometers

3. Crystals in enamel measure $\sim 50\text{ nm}$ not 100nm

4. Mastication forces in human teeth are not 1000N , more like max at 500N or lower.

5. Area in enamel depicted in Fig. 2 should be termed Hunter-Schreger band and shows change in alignment between large domains of enamel rods, which is visible in optical microscope (Fig. S4).

6. Page 7, line 251-254, authors discuss the lack of models describing the two mineral growth processes. This is not correct. There is an existing model that is widely accepted and was developed by Nanci et al in 1996. In this model the origin of the 2 different compartments, rod and interrod, of enamel crystals derives from an asymmetric deposition of matrix protein, which occurs at 2 distinct secretory sites a) at the distal end of the Tomes' process and b) through the proximal endings of the processes. This model fits well with the idea of amelogenin assembling into nanoribbons that template the growth of apatite along their backbone as described by Habelitz (JDR 2015), while the two papers referenced in this manuscript (Fang et al. 2011 and Tompson et al. 1978) appear not to provide any clues towards supporting control over distinguished crystal alignment in the two compartments interrod and rod enamel.

Overall, this manuscript contains some new information in particular with regards to the aprismatic enamel at the tooth surface and confirmatory information with regards to crystal orientation in interrod enamel. The manuscript would benefit from a re-evaluation of current description of enamel crystallites orientation and structure development in human enamel, in particular with regards to intra-rod enamel organization. The relevance of the MD simulation with regards to crack propagation across enamel rods is somewhat diminished without clinical or in vitro evidence beyond Fig S1D, which seems serendipitous.

Reviewer #3 (Remarks to the Author):

General Comments

The study reports on data which suggest that the c axes of crystals in the rod structures of dental enamel are not always co aligned with the crystal long axis. This would give rise to an overall structure which is highly resistant to crack propagation within and between crystals. This contrasts with interrod enamel where co-alignment does occur.

This is novel information achieved by use of a relatively novel technical approach (PIC).

The findings are of great interest to individuals in the enamel field but also to broader disciplines where physical properties of biological and non biological crystals are important.

There are a number of issues which the authors should consider mainly in terms of clarity.

This is potentially important data providing a new aspect of the structure of crystals in dental enamel. It is therefore important that the text is clear and unambiguous.

The main conclusion appears to be that the c axes of the apatite unit cells are not co aligned with the long axis of the crystals in enamel rods. This needs to be stated clearly.

In the text there is a confusion between 1. the orientation of the long axes of crystals and the long axes of the enamel rods in which they reside and 2. the orientation of the c- axes of unit cells within the crystals themselves. While this might be understood by workers in the enamel field, it is likely to be unclear to those not intimately involved in enamel research. The phrase "aligned and misorientated" (line 49) to distinguish these aspects of structure is unclear.

Some text revision is required here.

Detailed comments

Line 52 . 100 nm wide is rather high for enamel crystals. Species should be mentioned as in Figure S1

Line 101 The presumption of normal crystal growth in the c axis direction may be incorrect. For example data has been published indicating that enamel crystals exhibit regular morphological discontinuities. These relate to the formation of crystals from a series of regular subunits which fuse. Such fusion could easily give rise to crystallographic discontinuities, perhaps of the sort described in this paper. See:

Robinson C, Connell SD, Kirkham J, Shore RC , Smith A. J. Mater. Chem., 2004, 14, 2242 – 2248 2004)

Robinson C and Connell SD (doi: 10.3389/fphys.2017.00405 Frontiers in Physiology | www.frontiersin.org 1 June 2017 | Volume 8 | Article 405 (2017)

This data suggested that the crystals arise from the linear assembly of mineral matrix subunits which each nucleate mineral and allow the mineral units to fuse into the final crystal.

This origin of enamel crystals via a fusion of mineral /protein subunits could explain variations in the alignment of unit cells within each. This would depend on the nature of the subunits and how they orientated with respect to each another. For example, subunits could be arranged on occasion in a spiral manner:

Line 59 -60 This remark is too sweeping. A great deal has been published with regard to orientation of crystals in enamel. See Helmcke JG Ultrastructure of Enamel in Structural and Chemical Organisation of Teeth. 1967 Ac Press New York London 135-162. There are many more papers later than this, e.g. Daculsi, Kerebel.

This section also needs clarification. Are the authors here referring to the orientation of crystals and the orientation of unit cell axes within them?

Line 70. This sentence is ambiguous, " crystal orientation structure is unclear." What is hidden, is the fact that the c axis direction of the crystals is not co aligned with the long axis of the crystals at least in the main rod body.

Figure 1. Some images of crystal have shown them to be twisted in a spiral fashion. Would this affect the images seen in Figure 1. ?

Line 100 Why crystalline particles? Are these enamel crystals or not?

Line 116 What is meant by all polished similarly? Presumably they were all polished at the same time?

Why is 47 degrees much less effective in reducing crack propagation.

Line 214 Again assembly of crystals from subunits may provide an answer. Also the chemistry of the crystals should not be ignored. Interrod enamel is the first to be lost during carious attack and is known to be rich in carbonate .

Line 246. This statement presumes classical crystal initiation (nucleation) and growth. Fusion of matrix mineral subunits as referred to above could give rise to the differences between rod and interrod enamel depending on the nature of the subunits or simply how they are placed in the tissue.

Each subunit is an initiation/nucleation site and could give rise to differently orientated subunits within a final crystal.

Line 368. Enamel crystals are known to be defect lattices with calcium deficiencies, screw and point dislocations. These may be a result of, or related to, chemical differences. For example, the carbonate content of interrod enamel is likely to be higher than that of the rods themselves. In addition, the concentrations of carbonate and magnesium increase 4 - 6 fold in the direction from surface to interior. What might be the effect of such compositional differences on crystal behaviour?

If crystals are not orientated with their c-axes parallel, could the authors comment on the interface between such crystals, the disposition of ions at crystal surfaces, for example.

Responses to Reviewers' comments:

Reviewer #1 (Remarks to the Author):

We thank Reviewer #1 for excellent suggestions to improve the quality of the paper.

The authors applied innovative methods to take a new look at the microstructure of enamel crystallite organization in human teeth. The focus of the manuscript is on the application of X-ray circular dichroism analyses to discern between enamel crystallite shape arrangement, specifically the crystallite long axis, and the crystallographic c-axis of the carbonated hydroxyapatite crystallites in situ.

The paper questions the current status quo understanding in the field of enamel research that the long axis of enamel crystallites is co-aligned and co-oriented with the crystallographic c-axis. This is an interesting question indeed, however, the presented data do not support the claimed distinction between crystallite long axis and crystallographic c-axis. Consequently, the interpretations and molecular dynamics modeling, do not support the conclusion that the mis-orientation of crystallographic c-axes between the crystallites within a given enamel prism provides a mechanism to minimize crack propagation in the tooth crown.

We have added new data in Figure S2 page 23, to demonstrate that the crystals elongate parallel to one another in a rod, as in all other rods previously shown in the literature, whereas in PIC maps crystallites in each rod (a.k.a. head) are always differently oriented by $\sim 30^\circ$ in each area of the same size of that imaged in Figure S2B. These new data are now discussed on page 3, last paragraph.

The application of PIC mapping is novel for the analysis of human tooth enamel but has been explored and published in a recent publication in JACS (Stifler et al. 2018) using mouse enamel. The current understanding in the field is that murine enamel organization differs from human enamel in its arrangement of enamel rods, for example the decussation angle of rods is bigger in mouse than in human enamel. However, the fundamental process of tooth enamel formation has been shown to be conserved among mammals, including mineral phase, organic matrix components, crystallite size, shape, and organization into rods.

We thank the reviewer for acknowledging that the use of PIC mapping of human teeth is novel. We agree with the reviewer that many characteristics of enamel are common across mammals, yet there are significant differences in the structural organization of enamel in different species. This is especially true for murine incisors, which are highly specialized teeth, and their enamel has a unique structure. The Stifler *et al.* JACS paper showed the spectroscopy on which PIC mapping is based, and the applicability of PIC mapping to apatite in human bone and mouse enamel and dentin, thus it is completely distinct from this paper. The Stifler et al. reference, previously cited as “under review” is now published and thus updated. See ref. 30, page 17.

While the authors raise a very interesting question indeed,

Thank you

the presented evidence does not support the interpretation that the long axis of enamel crystallites is different from the crystallographic c-axis. This is mostly due to the lack of data such as SEM images at the same scale and location as PIC maps that would allow for the direct comparison between shape of crystals (alignment of crystal shape) and crystallographic c-axis seen in PIC mapping.

We have now show that the crystals within a rod are all aligned in the new Figure S2. As written in the caption this is representative of all rods. See page 23.

Importantly, that the structural organization of enamel crystallites into undulating prism and inter-prismatic enamel it is known and very well described for many species and illustrated in many publications. Based on this body or literature one would expect the crystallite axis and alignment between crystallites to change continuously, that is to go in and out of plane of sectioning, a pattern known as Hunter-Schreger-bands, as well as to change their orientation within prisms.

The reviewer is right, the elongation direction of the rods changes going in and out of plane and thus creating the decussating pattern also known as Hunter-Schreger-Bands (e.g. Figure 2). The new observation here is that *within a rod* (head) there is a significant degree of misorientation, and neighboring crystals at the nanoscale differ in orientations by 1°-30°. This novel observation challenges the consensus opinion in the field that c-axes of neighboring crystals in enamel rods are co-aligned (see ref. 15: Travis & Glimcher, JCB, 1964, and ref. 14: Glimcher et al. J. Ultrastr. Res., 1965). Importantly the difference in the c-axes orientation are significantly greater than the misorientation of the long axes of the crystals (Figure S1, and new Figure S2 on page 23). We state now much more clearly that the misorientation is *within the rods*. See summary in page 1, findings on page 3, 1st paragraph, and that this observation cannot be due to the rods changing direction, as the reviewer suggests. See page 5, 2nd paragraph after the figure caption.

A body of literature describes the hierarchical arrangement of tooth enamel and enamel crystal dimensions and phase.

We previously cited several of these papers and have now cited them all.

For example: crystallite dimensions with thickness in the 20nm range, width being below 100nm (Selvig 1972)
(Kerebel et al. 1979)
(Daculsi et al. 1978, Daculsi et al. 1984)

See refs 18-21.

Fracture behavior of enamel under consideration of crystallite alignment and enamel prism organization (Scheider et al. 2015, Yilmaz et al. 2015, Yilmaz et al. 2015).

See text on pages 8, 1st paragraph and refs 62-65.

Synchrotron analyses tracking enamel prisms from the DEJ to the enamel surface to show the cork-screw paths of enamel rods. The implication is that at any plane of section the enamel prisms, especially in the zone of inner enamel, and crystallites within the prism will be in plane for only a short path length and undulate in and out of plane of section (Tafforeau et al. 2012).

This and other synchrotron studies are now cited on page 2 and refs 23-26.

SEM data illustrating bovine enamel microstructure, including decussation pattern of prism and crystallites within prisms not being in a simple parallel arrangement but organized similar to the twisting fibers in a thread of wool (Wang et al. 2012).

See new text on page 2, 2nd paragraph and refs 30, 31.

SEM images showing human enamel microstructure, including decussation pattern and Hunter Schreger Bands, that is the undulating of enamel prisms (Risnes 1998).

See text on page 2 and ref 23.

Methods description and application of linear dichroism to apatite, in particular mouse enamel prisms and crystal arrangement in mouse enamel (Stifler et al. 2018)

This paper is now published and thus its reference was updated, and newly cited in the introduction regarding mouse enamel and in the methods. Page 2, 3rd paragraph, page 11, 2nd paragraph, and ref 30.

Relevance and effects of hydration status of enamel and protein content on the mechanical properties (Baldassari et al 2008).

See text on page 8, 1st paragraph and ref 66.

The following further evidence would be required to strengthen the conclusions.

- SEM images at the same scale and ideally the same location as the PIC mapping figures to show the arrangement of crystallites and that the crystallite long axis does not coincide with crystal c-axis orientation

We have acquired new SEM images of etched enamel on 3 different human teeth and many different rods in each sample, all similar to one another, and all showing parallel elongated crystallites within each rod. We tried but it was not possible to identify precisely the same location as in Figure 2, as described in page 12, 2nd paragraph. This is not particularly relevant, however, because all rods look the same after etching, with parallel elongated crystallites, and all rods look the same in every PIC map, with gradually varying c-axis orientations. The new SEM data are presented in new Figure S2 page 23, and the fact that they are representative is described in Figure S2 caption, page 23.

- TEM sections or FIB-lift outs that are either thinner than 100 nm to not include several crystallites superimposed, or are at 100 nm thickness and include single crystallites

We tried, but producing FIB sections thinner than 100 nm turned out to be impossible: they fall apart and there is nothing left to image in the TEM that is recognizably crystalline. The section in Figure S4 is 100 nm thick. Considering this size, they are all nearby and run parallel to one another, as shown in Figures S4, S2 and in many papers before ours. Thus, the observation in Figure S4 that crystallites that appear parallel to one another are 23° , 27° and $\geq 18^\circ$ apart in orientation demonstrates our new finding. We now explicitly explain this on page 4, last paragraph.

In addition, the location of the FIB section is now shown in Figure S5, page 26.

- The thickness of enamel crystallites is in the 20 nm range (Kerebel et al. 1979), and the TEM images in the present manuscript show superimposed apatite crystals. The diffraction patterns are, therefore not of single crystallites.

The reviewer is right, the thickness of enamel crystals is indeed less than 100 nm. They are on average 26 nm x 70 nm. This, however, does not preclude us from studying the alignment of neighboring crystals. We now explained that the image represents a volume. Considering the average crystallite sizes, in this volume there could be 5-10 crystallites, contributing to the FFT power spectrum in Figure S4B. Yet the angle spread of c-axes based on the FFT power spectrum is 27° . Further analysis of smaller regions of adjacent crystals (Figure S4C,D) revealed that the c-axes of adjacent crystals are more than 20° apart. We have now indexed the power spectra in Figure S4 and rewrote the text and the figure caption to clarify these points. See page 4, last paragraph, and Figure S4 caption in pages 25-26.

In addition, the diffraction pattern C does not show the clear orientation as indicated by the arrow in Figure S3.

- Methods should provide an adequate protocol for sample preparation,

The Methods section now provides many more details on sample preparation. See pages 10-12.

although it is known that dehydration and loss of organic material affects the mechanical properties of tooth enamel (Baldassarri et al 2008).

We did not remove organics from the samples and although the samples were dehydrated, it is highly unlikely that the dehydration could lead to changes in the structural organization of the crystals. We have previously demonstrated that dehydrated and later rehydrated enamel has the same mechanical properties as enamel which was never dehydrated (Baldassarri et al. 2008, now cited as ref. 66, discussed on page 8, 1st paragraph).

- The methods need to include which teeth were actually used, whether they were extracted for orthodontic reasons and with IRB approval, whether the enamel had any defects, whether they

had been stored and how, were fixed and/or dehydrated, what the exact position and plane of sectioning was.

These and many other important details were added to the methods on page 10.

- The authors state, based on their molecular dynamics modelling, that mis-orientation of adjacent crystallites induces crack deflection. Yet, the model ignores the fact the interface between crystallites is not perfect and contains protein and water.

The reviewer is right, there are occasionally water and proteins at crystal interfaces, but they are not ubiquitous, and they were omitted on purpose from our simulations. It is well known that at heterogeneous materials interfaces cracks are deflected. The new discovery here is that at interfaces of *the same material* cracks are deflected, provided the crystals are differently oriented. We have clarified this point on page 6, 1st paragraph after the figure.

- The modelling approach and data interpretation needs not acknowledge that the presence of lattice defects and substitutions can affect the crystallite structure.

We agree and would be glad to remove it, but we cannot find where this was mentioned.

Several publications provide in depth explorations of the mechanical properties of tooth enamel under consideration of prisms and crystallite orientation and presented experimental data (Scheider et al. 2015, Yilmaz et al. 2015, Yilmaz et al. 2015). These studies have more immediate and biological relevance than the modelling data provided in the presented manuscript, that ignore presence of water and organic molecules between crystallites.

We thank the reviewer for pointing out these excellent papers. We have now commented on them on page 8, 1st paragraph and cited them as refs 62-65.

Bibliography;

- Baldassarri M, Margolis HC, Beniash E. 2008. Compositional determinants of mechanical properties of enamel. *J Dent Res.* Jul;87(7):645-9
- Daculsi G, Kerebel B. 1978. High-resolution electron microscope study of human enamel crystallites: size, shape, and growth. *J Ultrastruct Res.* 65(2):163-72.
- Daculsi G, Menanteau J, LM; K, D; M. 1984. Length and shape of enamel crystals. *Calcif Tissue Int.* 36(5):550-5.
- Kerebel B, Daculsi G, Kerebel LM. 1979. Ultrastructural studies of enamel crystallites. *J Dent Res.* 58(Spec Issue B):844-51.
- Risnes S. 1998. Growth tracks in dental enamel. *J Hum Evol.* 35(4-5):331-50.
- Scheider I, Xiao T, Yilmaz E, Schneider GA, Huber N, Bargmann S. 2015. Damage modeling of small-scale experiments on dental enamel with hierarchical microstructure. *Acta Biomater.* 15:244-53.
- Selvig KA. 1972. The crystal structure of hydroxyapatite in dental enamel as seen with the electron microscope. *J Ultrastruct Res.* 41(3):369-75.

Stifler CA, Wittig NK, Sassi M, Sun CY, Marcus MA, Birkedal H, Beniash E, Rosso KM, Gilbert P. 2018. X-ray Linear Dichroism in Apatite. *J Am Chem Soc.* 140(37):11698-704.

Tafforeau P, Zermeno JP, Smith TM. 2012. Tracking cellular-level enamel growth and structure in 4D with synchrotron imaging. *J Hum Evol.* 62(3):424-8.

Wang C, Li Y, Wang X, Zhang L, Tiantang, Fu B. 2012. The enamel microstructures of bovine mandibular incisors. *Anat Rec (Hoboken).* 295(10):1698-706.

Yilmaz ED, Jelitto H, Schneider GA. 2015. Uniaxial compressive behavior of micro-pillars of dental enamel characterized in multiple directions. *Acta Biomater.* 16:187-95.

Yilmaz ED, Schneider GA, Swain MV. 2015. Influence of structural hierarchy on the fracture behaviour of tooth enamel. *Philos Trans A Math Phys Eng Sci.* 373(2038).

All these references are now discussed and cited as detailed above.

Reviewer #2 (Remarks to the Author):

This paper reports on observations of the microstructure of human dental enamel using a novel analysis technique called PIC-PEEM. This technique allows the detection of crystal orientation in association with EM imaging over larger areas. The authors state that they discovered new structural elements related to the alignment of fibrous apatite crystals in enamel: A) continuous alignment of fibrous apatite crystal in the interrod enamel; B) mis-orientation and gradual change of apatite orientation within an enamel rod; C) aprismatic enamel at the occlusal surface is not comprised of apatite fibers perpendicular to the surface. In addition MD simulations were performed to demonstrate the role of mis-alignment of crystals in crack propagation.

Some of the information the authors provided is novel. In particular observation C runs contrary to what is mostly stated in the literature. The current view is that the outer aprismatic enamel has apatite crystals oriented in parallel with the c-axis perpendicular to the occlusal surface. In the PIC map provided in Fig 4, the aprismatic enamel has the same orientation as the interrod enamel which is intriguing and plausible. This is a significant novel observation.

Thank you.

To ensure any artifact it would be beneficial to obtain a TEM or SEM image of the surface zone and demonstrate the alignment of the enamel crystallites with regards to the occlusal surface.

We agree that ruling out artifact is desirable, but in this case we are confident that there are no artifacts, in the measurement of the c-axis orientation by PIC mapping, as confirmed by rotating the same sample by 90° in Figure S6. We tried to etch and then image precisely the same regions analyzed by PIC mapping in aprismatic enamel, but did not obtain any useful results. The sample at first was not etched enough, then it was etched too much, thus the regions are unrecognizable. There is ample literature, however, demonstrating that the aprismatic enamel crystallites run perpendicular to the surface of the tooth, thus we do not deem strictly necessary to further confirm this point in our samples. See for example Ten Cate's book, 9th edition 2017, Fig. 7-38, page 333.

Observation A, refers to the interrod enamel. Interrod enamel has long been speculated to be a continuous phase and many SEM images have shown such an alignment, see work by Warshawsky in the 70s and more recent work by Shane White, the Ariola group and Schneider group on fracture mechanics (see refs in the manuscript). The PIC map of Figure 4 adds additional information and more clearly demonstrates the alignment of apatite fibers through the thickness of the enamel and confirms the term continuous phase used by previous authors.

We agree with the reviewer that earlier SEM observations suggest that the crystals in the interrod matrix might be co-aligned based on their appearance. However, SEM cannot provide crystallographic information on apatites. Here, for the first time, we directly demonstrate that the crystal c-axes are co-oriented in the interrod mineral over large areas of enamel. As suggested by the reviewer we now write: "Interrod enamel has long been speculated to be a continuous phase based on the alignment of its apatite fibers observed in SEM images^{13,70-73}. The results of Figures 2, 5, S3, S5, S6 show that not only are the interrod crystallites aligned, but their c-axes

are highly co-oriented. This confirms that the term “continuous phase” used by previous authors¹³ for the interrod was accurate.” See page 8, 4th paragraph, and new refs 70-73.

Observation B, was surprisingly characterized as a novel observation. While certainly in simplified models that describe the mechanical behavior of enamel a parallel alignment within rods has been used to ease simulation calculations, it is widely reported that apatite fibers in enamel rods are not aligned in parallel across an enamel rod. In fact most current oral/dental histology textbooks still use the illustration by Meckel from 1965 (Arch Oral Biol.) to illustrate the change in orientation from head to tail within a keyhole-like structure (see Ten Cate textbook referenced in manuscript), see also Boyde 1967. It is surprising that this existing model was overlooked by the authors. While Meckel did not believe in the presence of interrod enamel, which is incorrect in his model, he correctly described the lateral flare of apatite crystals across enamel rods which has been shown in many studies involving SEM analysis.

We agree with the reviewer that there is a gradual transition between the head (rod) and tail (interrod) enamel, as reported by all of the above papers and textbook. Our maps clearly show that while the boundaries between the rod and adjacent interrod are sharp, the transition between the rod and interrod within the same “keyhole” unit is gradual, in excellent agreement with the previous model. See Figures 1, 2 or S5 for example. What we are focusing on is the crystallographic misalignment in each rod (head), which contradicts the previous paradigm, mostly based on SAED diffraction studies. We have now clarified that we meant “within each rod” in the text. See summary page 1, page 2, page 5, 2nd paragraph after figure, page 8, 1st paragraph, Figure S2 caption page 23. The above references are now discussed on page 3 and page 5, and cited as refs 46, 49, 50.

A benefit of this orientation to crack deflection is somewhat obvious, but has, to this reviewer’s knowledge, never been considered previously.

We thank the reviewer for acknowledging the novelty of this toughening mechanism. We don’t think it is obvious at all. It was neither predicted, predictable, observed, nor discussed by anyone else, thus it does not seem obvious to us.

However, clinically and in vitro observations clearly suggests that cracks follow the long axis of the enamel rod and run through the interrod-rod interface in enamel. An analysis of the effect of the about 30 degree oblique angle of interrod enamel to crack propagation appears more relevant to our understanding of cracking in enamel.

Regarding cracks following the long axis of the enamel rod and running through the interrod-rod interface in enamel, we point out that at this interface there is an organic sheath, labeled S in Figure 1 and discussed in the above cited reference (Meckel 1965). Cracking deflection and propagation at materials interfaces is a well-known phenomenon, observed in sponge spicules (Miserez, Weaver et al. Adv Funct Mater 2008) and self-sharpening sea urchin teeth (Killian, Metzler et al. Adv Funct Mater 2011), among many other systems. The crack deflections described here are at interfaces of identical materials, differing only in orientation, which is a novel mechanism. This is now mentioned in the text. See page 6, 1st paragraph after figure, and refs 42, 54.

The fact that extensive fractures are never observed within a rod proves our point. This is now mentioned in the text. See page 8, 2nd paragraph and ref 63.

Also, of concern in the MD simulations of cracks within a rod is the lack of evidence of fine enough crack radii that actually are able to interact between the enamel crystallites. In this regards the statements that enamel crystallites measure ~100nm is not correct. It is well documented that crystal measure about ~50nm in cross sections (40-60nm).

We thank the reviewer for pointing this out. We have corrected the inaccurate statement, and provided a reference for the well-known ~50 nm size of the crystallites. See page 1, 1st paragraph, page 2, 1st paragraph, page 3, Figure 1 caption, page 23 Figure S2 caption and Daculsi and Kerebel ref 18. It is noted that the layer height of the model for the FEM is not crucial to our result, as we have tested heights from 50 nm to 150 nm by keeping all the other FEM setting the same and the strain field within tens of nm of the crack tip is not significantly affected by the crystal thickness. Moreover, the crystal thickness parameter is only used within the FEM to accurately compute the profile of the strain field within the crack tip, which is then applied as the boundary condition for the full atomistic model. The full atomistic model provides the finest description of the interaction between the crack tip and the crystal interface within tens of nm and its result will not be affected by this thickness change. This is now explained on page 14.

Therefore the assignment in Fig. 3C of changing crystal orientations at domains of about 100nm is not due to single crystal mis-alignments.

We think the reviewer is referring to Figure 1C not 3C, where we showed 100 nm ticks on a ruler, but domains of single color are larger, clearly indicating more than one crystallite in each co-oriented homo-colored domain. We have removed the confusing ruler, and explained this point in the text more clearly. See Figure 1 caption, page 3.

Details:

1. Information about the thickness of the two specimens used for PIC-PEEM analysis was not provided.

The thickness is 3 mm, however PIC mapping is a surface method that probes the top 3 nm of the polished and coated surface, as we now explain in the methods. See page 10 Samples.

In this regards it would be of keen interests to provide information about the penetration depth of the X-ray analysis and its impact on the data collected was not provided, but are critical to the interpretation of the findings. Also some description of the method used would be useful to the reader.

We added this description on page 11, 4th paragraph.

3. Crystals in enamel measure ~50 nm not 100nm

The crystal size of mature enamel in humans is 26 nm x 63 nm according to Daculsi & Kerebel 1978. We have corrected this in the text on page 1, page 2, 1st paragraph, Figure 1 caption page

3, and Figure S2 caption page 23.

4. Mastication forces in human teeth are not 1000N, more like max at 500N or lower.

The reviewer is absolutely right, and we thank her/him for catching this mistake. Typical masticatory forces are around 100 N. They can reach maximal values of 770 N in young males, according to Varga *et al.*, even though Wegst *et al.* rounded it up to 1000 N, which is the source we used before. We corrected the statement in the introduction and in the results sections. See page 2, 1st paragraph, bottom of page 7, and refs 1 and 2.

5. Area in enamel depicted in Fig. 2 should be termed Hunter-Schreger band and shows change in alignment between large domains of enamel rods, which is visible in optical microscope (Fig. S4).

Agreed, we mentioned the “Hunter-Schreger bands or decussation pattern” once in the text on page 2 paragraph 1 and in Figure 2 caption page 5. This is useful to define once, as it is widely used in the older literature, but in the rest of the text we prefer to use decussation pattern rather than HSB, as it is the modern term.

6. Page 7, line 251-254, authors discuss the lack of models describing the two mineral growth processes. This is not correct. There is an existing model that is widely accepted and was developed by Nanci *et al.* in 1996. In this model the origin of the 2 different compartments, rod and interrod, of enamel crystals derives from an asymmetric deposition of matrix protein, which occurs at 2 distinct secretory sites a) at the distal end of the Tomes’ process and b) through the proximal endings of the processes. This model fits well with the idea of amelogenin assembling into nanoribbons that template the growth of apatite along their backbone as described by Habelitz (JDR 2015), while the two papers referenced in this manuscript (Fang *et al.* 2011 and Tompson *et al.* 1978) appear not to provide any clues towards supporting control over distinguished crystal alignment in the two compartments interrod and rod enamel.

This is a great idea, for which we thank the reviewer, and which we have included as a third possibility. See page 9 and ref 78.

Overall, this manuscript contains some new information in particular with regards to the aprismatic enamel at the tooth surface and confirmatory information with regards to crystal orientation in interrod enamel. The manuscript would benefit from a re-evaluation of current description of enamel crystallites orientation and structure development in human enamel, in particular with regards to intra-rod enamel organization. The relevance of the MD simulation with regards to crack propagation across enamel rods is somewhat diminished without clinical or *in vitro* evidence beyond Fig S1D, which seems serendipitous.

We respectfully disagree: the fact that crystal mis-orientations provide a toughening mechanism is new and exciting. MD simulations demonstrate that the concept, first formulated as a cartoon, is indeed sound. Future experiments, done by us or, even better, by researcher far more expert than us in fracture experiments and quantitative measurements, will correlate mis-orientations and toughening and thus test the hypothesis formulated in this paper. One of us (PG) recently

wrote a proposal for a 3-year NSF grant that will put this hypothesis through a rigorous and quantitative test. This is well beyond the scope of the present manuscript, which is exclusively about mis-orientation discovery.

We now say “The fact that in human teeth fractures are not observed across rods, but always at the micro-scale interrod-rod interface⁶³ demonstrates that the nano-scale toughening mechanisms proposed here is effective.” See page 8, 2nd paragraph.

We thank Reviewer #2 for this and all other excellent suggestions, which improved the quality of the paper.

Reviewer #3 (Remarks to the Author):

We thank Reviewer #3 for all the useful suggestions, resulting in a higher quality and clearer paper.

General Comments

The study reports on data which suggest that the c axes of crystals in the rod structures of dental enamel are not always co aligned with the crystal long axis. This would give rise to an overall structure which is highly resistant to crack propagation within and between crystals. This contrasts with interrod enamel where co-alignment does occur.

This is novel information achieved by use of a relatively novel technical approach (PIC).

The findings are of great interest to individuals in the enamel field but also to broader disciplines where physical properties of biological and non biological crystals are important.

Thank you

There are a number of issues which the authors should consider mainly in terms of clarity.

This is potentially important data providing a new aspect of the structure of crystals in dental enamel. It is therefore important that the text is clear and unambiguous.

We agree, and have made all the changes as described below.

The main conclusion appears to be that the c axes of the apatite unit cells are not co aligned with the long axis of the crystals in enamel rods. This needs to be stated clearly.

We have now stated this clearly in the summary and the findings, and shown new data in Figure S2. See summary page 1, page 3, 1st paragraph, page 5, 2nd paragraph after the figure, page 8, 3rd paragraph, Figure S2 caption page 23, Figure S4 caption page 25.

In the text there is a confusion between 1. the orientation of the long axes of crystals and the long axes of the enamel rods in which they reside and 2. the orientation of the c- axes of unit cells within the crystals themselves. While this might be understood by workers in the enamel field, it is likely to be unclear to those not intimately involved in enamel research. The phrase "aligned and misorientated" (line 49) to distinguish these aspects of structure is unclear. Some text revision is required here.

We revised and clarified how we use the words aligned and oriented. See page 2, 1st paragraph.

Detailed comments

Line 52 . 100 nm wide is rather high for enamel crystals. Species should be mentioned as in Figure S1

We agree with the reviewer, the crystal size of mature enamel in humans is 26 nm x 63 nm according to Daculsi & Kerebel 1978. We have corrected the size and now call it ~50 nm in the text on page 1, page 2, 1st paragraph, Figure 1 caption page 3, and Figure S2 caption page 23.

Line 101 The presumption of normal crystal growth in the c axis direction may be incorrect. For example data has been published indicating that enamel crystals exhibit regular morphological discontinuities. These relate to the formation of crystals from a series of regular subunits which fuse. Such fusion could easily give rise to crystallographic discontinuities, perhaps of the sort described in this paper. See:

Robinson C, Connell SD, Kirkham J, Shore RC, Smith A. J. Mater. Chem., 2004, 14, 2242 – 2248 2004)

Robinson C and Connell SD (doi: 10.3389/fphys.2017.00405 Frontiers in Physiology | www.frontiersin.org 1 June 2017 | Volume 8 | Article 405 (2017)

This data suggested that the crystals arise from the linear assembly of mineral matrix subunits which each nucleate mineral and allow the mineral units to fuse into the final crystal. This origin of enamel crystals via a fusion of mineral /protein subunits could explain variations in the alignment of unit cells within each. This would depend on the nature of the subunits and how they orientated with respect to each another. For example, subunits could be arranged on occasion in a spiral manner:

This is a great idea, for which we thank the reviewer. We have now included this idea in the 3rd paragraph of page 8, 3rd paragraph and refs 67-69.

Line 59 -60 This remark is too sweeping. A great deal has been published with regard to orientation of crystals in enamel. See Helmcke JG Ultrastructure of Enamel in Structural and Chemical Organisation of Teeth. 1967 Ac Press New York London 135-162. There are many more papers later than this, e.g. Daculsi, Kerebel.

We cited the two refs 18, 32 in page 2. We also clarified that “very little is known regarding how crystals are oriented *within* this organization, especially at the scale of tens or hundreds of microns” on page 2, last paragraph.

This section also needs clarification. Are the authors here referring to the orientation of crystals and the orientation of unit cell axes within them?

We have now clarified in the 1st paragraph of page 2 that “we use the word “aligned” when referring to morphological alignment of elongated and parallel crystals, and “mis- oriented” or “co-oriented” when referring to the orientation of crystalline *c*-axes.” This should remove any ambiguity throughout the text.

Line 70. This sentence is ambiguous, " crystal orientation structure is unclear." What is hidden, is the fact that the c axis direction of the crystals is not co aligned with the long axis of the crystals at least in the main rod body.

We simplified and clarified the statement. See page 2, last paragraph.

Figure 1. Some images of crystal have shown them to be twisted in a spiral fashion. Would this affect the images seen in Figure 1. ?

In bovine enamel indeed crystal elongations spiral along the long axis of the rod. Whether or not *c*-axes also spiral we don't know. PIC mapping would reveal it, but has never been used in bovine enamel. Here in human enamel we do not observe spiraling patterns in those rods imaged in circular cross-section, e.g. those that vary from red to black near the center of Figure 2. Nothing in the data we observed thus far suggests a spiraling pattern of crystal orientations.

Line 100 Why crystalline particles? Are these enamel crystals or not?

We changed it to "TEM studies found crystals in the enamel rods" page 3, last paragraph.

Line 116 What is meant by all polished similarly? Presumably they were all polished at the same time?

We changed it to "In the present PIC maps all crystals are simply polished, hence their orientations are expected to remain as they were in pristine enamel." See middle of page 4.

Why is 47 degrees much less effective in reducing crack propagation.

Interesting question, which surprised us as well, but it is reproducible and believable. We now write "This result was reproduced in multiple simulations, using homogenous and inhomogeneous loading in the horizontal direction, and is therefore noteworthy, even though it was unexpected." See page 6, near the top. Small angles appear to be more effective at crack deflection than large angles. We thus added a comment about this on page 6, 2nd to last paragraph, and added new Figure 4 to show this interesting and important result.

Line 214 Again assembly of crystals from subunits may provide an answer. Also the chemistry of the crystals should not be ignored. Interrod enamel is the first to be lost during carious attack and is known to be rich in carbonate .

Agreed. But no changes were made as this will be subject of future studies by other groups. We do not think we will pursue this idea, thus if the reviewer is interested, she/he will be free to pursue her/his idea and substantiate it with data.

Line 246. This statement presumes classical crystal initiation (nucleation) and growth. Fusion of matrix mineral subunits as referred to above could give rise to the differences between rod and interrod enamel depending on the nature of the subunits or simply how they are placed in the tissue. Each subunit is an initiation/nucleation site and could give rise to differently orientated subunits within a final crystal.

We agree, and have commented about fusion about this on page 8, 3rd paragraph.

Line 368. Enamel crystals are known to be defect lattices with calcium deficiencies, screw and point dislocations. These may be a result of, or related to, chemical differences. For example, the

carbonate content of interrod enamel is likely to be higher than that of the rods themselves. In addition, the concentrations of carbonate and magnesium increase 4 - 6 fold in the direction from surface to interior. What might be the effect of such compositional differences on crystal behaviour?

We have no idea, but it is a good question, which deserves an answer in the future, beyond the scope of this paper.

If crystals are not orientated with their c-axes parallel , could the authors comment on the interface between such crystals, the disposition of ions at crystal surfaces, for example.

No, we cannot. This is why we did MD simulations, as they are far better than any speculation. We find it surprising and fascinating that mis-oriented crystals under realistic masticatory pressures sinter, and thus fuse into a single crystal, albeit with different orientations. We have commented about this on page 8, 3rd paragraph.

Reviewers' comments:

Reviewer #1 (Remarks to the Author):

The revised manuscript incorporates a lot of additional and helpful information, especially in the methods part, as well as helpful clarifications.

As for Methods description, it is rather surprising that EDTA etching should have occurred after Pt-coating, as described now on page 10, line 367, 368.

A few fundamental questions of the manuscript remain not really resolved, though, as listed below.

Original critique: ...the presented evidence does not support the interpretation that the long axis of enamel crystallites is different from the crystallographic c-axis. This is mostly due to the lack of data such as SEM images at the same scale and location as PIC maps that would allow for the direct comparison between shape of crystals (alignment of crystal shape) and crystallographic c-axis seen in PIC mapping.

Rebuttal: We have now show that the crystals within a rod are all aligned in the new Figure S2. As written in the caption this is representative of all rods. See page 23.

"representative of all rods" is a rather strong statement, especially given that within any given tooth prism angles change and it is known that within the prism the crystallite packing and alignment varies from DEJ to surface.

The images provided in Figure S2 are not next to the PIC maps, but in the supplemental information, making a direct comparison between morphological c-axis arrangement in crystals somewhat cumbersome. Furthermore, the additional SEM images are also different scales compared to the PIC maps, that is about twice the field of view for the low magnification and less than half the field of view for the higher magnification images. Nevertheless, the Figure S2 B and D clearly show that crystallite morphological alignment does vary and is not strictly parallel, as the authors claim so emphatically. If the strictly parallel arrangement was so obvious, an SEM image of the same magnification and size of field of view next to the PIC map would make that point. But that information is still missing and hence, the argument is not convincing, as variability in crystallite alignment is clearly visible within both prisms and interprismatic enamel.

Critique: Importantly, that the structural organization of enamel crystallites into undulating prism and inter-prismatic enamel it is known and very well described for many species and illustrated in many publications. Based on this body or literature one would expect the crystallite axis and alignment between crystallites to change continuously, that is to go in and out of plane of sectioning, a pattern known as Hunter-Schreger-bands, as well as to change their orientation within prisms.

Rebuttal: The reviewer is right, the elongation direction of the rods changes going in and out of plane and thus creating the decussating pattern also known as Hunter-Schreger-Bands (e.g. Figure 2). The new observation here is that within a rod (head) there is a significant degree of misorientation, and neighboring crystals at the nanoscale differ in orientations by 1°-30°. This novel observation challenges the consensus opinion in the field that c-axes of neighboring crystals in enamel rods are co-aligned (see ref. 15: Travis & Glimcher, JCB, 1964, and ref. 14: Glimcher et al. J. Ultrastr. Res., 1965). Importantly the difference in the c-axes orientation are significantly greater than the misorientation of the long axes of the crystals (Figure S1, and new Figure S2 on page 23). We state now much more clearly that the misorientation is within the rods. See summary in page 1, findings on page 3, 1st paragraph, and that this observation cannot be due to the rods changing direction, as the reviewer suggests. See page 5, 2nd paragraph after the figure caption.

The authors do state their interpretations now much more clearly. The PIC maps are impressive and beautiful and maybe there could be clear evidence in Figures S5 and S6 that crystallite long axis and crystallographic c-axis differ. A SEM image simply showing crystallite morphology in the ca. 30 um aprismatic enamel layer might drive the point home and remove the doubt and counter argument that crystallites within prismatic enamel are perpendicular to the crown surface and not necessarily perpendicular to the edge of the section, which may or may not be slightly out of plane.

Critique:

...although it is known that dehydration and loss of organic material affects the mechanical properties of tooth enamel (Baldassarri et al 2008).

Rebuttal: We have previously demonstrated that dehydrated and later rehydrated enamel has the same mechanical properties as enamel which was never dehydrated (Baldassarri et al. 2008, now cited as ref. 66, discussed on page 8, 1st paragraph).

The rebuttal argument taken from Baldassarri et al. refers to the sample processing procedure and the point that a freeze-dried sample can be rehydrated and that mechanical properties are the same before freeze drying and after re-hydrating. Whereas all the experimental data of the Baldassarri et al. reference show that samples without water or without organic material have very different properties compared to samples that are wet or with organic material. These data and this argument should be considered for the MD modeling the neglects both organic material and a hydration layer between crystallites.

Critique:

- The modeling approach and data interpretation needs not acknowledge that the presence of lattice defects and substitutions can affect the crystallite structure.

Rebuttal: We agree and would be glad to remove it, but we cannot find where this was mentioned.

The rebuttal is, unfortunately, focusing on the typo of "not" which would correctly be "needs _to_" acknowledge...

I apologize for the typo, but because this is a very critical point it _does need to be addressed_ in the manuscript.

- The modeling approach and data interpretation needs to acknowledge that the presence of lattice defects and substitutions can affect the crystallite structure.

Overall, the paper shows truly beautiful PIC maps and electron microscopy data but leaves the reader wondering why the authors are forcing the point of discerning between morphological long axis and crystallographic c-axis, when the parsimonious explanation would be that the controlled mismatch in crystal long axis alignment optimizes the mechanical properties of enamel.

Reviewer #2 (Remarks to the Author):

The authors have provided an extensive rebuttal to the initial review. References are now updated and the list was extended appropriately. Overall however there were many additions to the supplemental portion of this study, but little change to the original material or data as provided in the first submission was made.

As mentioned in the critiques of the first review, a single enamel rod shows a change in fiber orientation from a more parallel orientation of the fibers' long axes to the rod axes to an angle of

about 30 degree as laid out by Meckel and others in the 1960. The authors responded to this comment with: "What we are focusing on is the crystallographic misalignment in each rod (head), which contradicts the previous paradigm, mostly based on SAED diffraction studies." This is not true. Meckel's schematic shows exactly this and change in orientation with a single rod! In addition, the authors continued to state: "This means that, contrary to earlier reports, the long axis of each nanocrystal is not necessarily co-aligned with the crystalline c-axis, they can be as much as 60° apart." There seems to be a misconception: Apparently SEM, TEM and AFM all show that fibers flare laterally towards the tail of a key-hole shaped enamel rod. Therefore the c-axis of these fibers will also flare, which is what is shown in the PIC maps of Fig 1. It is not clear why the authors need to make a distinction and claim that the crystal long axis does not correspond to the crystallographic c-axis.

If in any case this was supposedly true, a full correlation of SEM image and the identical PIC-Map needs to be presented. Therefore the main criticism remained, there is no data provided that shows a clear relationship between the microstructure of enamel, e.g. nanofiber orientation shown by SEM (or TEM) and the crystallographic orientation suggested by PIC mapping. Correlated images need to be provided to support the hypothesis that the crystal long axes does not correspond with the crystallographic c-axis. The PIC map of figure 2 is impressive but needs support from EM data. The main observation of a continuous phase of apatite crystals in the interrod enamel remains and is very interesting and a clear proof of models that others have suggested without such strong evidence, mainly relying on EM evidence. PIC maps facilitate a more extended view of the interrod orientation. However extended crystal orientation maps using SEM and TEM correlated with EBSD were recent shown by Koblischka-Veneva et al., Nano Research 2018.

Furthermore, MD simulation continues to lack a support by any data that cracks would penetrate enamel in the fashion described.

Reviewer #3 (Remarks to the Author):

The authors have gone to considerable lengths to both answer and take on board queries and suggestions made by the reviewers. Additional material and amendments have been added in this respect. The paper can be published.

Response to Reviewers' comments:

Reviewer #1 (Remarks to the Author):

The revised manuscript incorporates a lot of additional and helpful information, especially in the methods part, as well as helpful clarifications.

Thank you!

As for Methods description, it is rather surprising that EDTA etching should have occurred after Pt-coating, as described now on page 10, line 367, 368.

The reviewer is right, etching occurred unevenly, and only where the coating got peeled off. For this reason we have now adopted a better, and well-established etching method: 2 seconds in 10 v% HCl, which gave much better, and more reproducible results. This method uses more concentrated HCl and less time (6 mol% HCl for 2 sec) than Habelitz 2001 used for AFM imaging (0.05 mol% HCl for 20 sec), but it is a common technique for etching fossil teeth for SEM imaging (e.g. Botella 2008, Gillis 2007, and Reif 1974). In the literature, acid etching of human teeth has been optimized to improve bonding for dental implants/coatings, not for SEM imaging. In dentistry, the prevailing technique is to etch in 30% phosphoric acid for about 15 sec (e.g. Silverstone 1975, Kodaka 1993, and Shinohara 2006). The etching recipe typically used for more fragile fossil teeth was selected to reduce damage to our tooth sample. See page 12 for the additional methods: "The human molar shown in Figures 3, S4, S5, and S13 was gently polished using 50 nm alumina suspension (Masterprep, Buehler, Lake Bluff, IL) saturated with CaCl_2 to remove the Pt coating. Following common protocol for etching fossil teeth for SEM imaging⁸⁶⁻⁸⁸, the tooth was then etched in 10 (v/v)% HCl for 1 second, rinsed twice in DD-H₂O adjusted to pH 8, once in pure ethanol, and dried with dry CO₂ and coated as described below. However, 1-second etching was not sufficient for quality SEM imaging, so the tooth was gently polished again, and etched for 1 additional second as described above."

A few fundamental questions of the manuscript remain not really resolved, though, as listed below.

Original critique: ...the presented evidence does not support the interpretation that the long axis of enamel crystallites is different from the crystallographic c-axis. This is mostly due to the lack of data such as SEM images at the same scale and location as PIC maps that would allow for the direct comparison between shape of crystals (alignment of crystal shape) and crystallographic c-axis seen in PIC mapping.

Rebuttal: We have now show that the crystals within a rod are all aligned in the new Figure S2. As written in the caption this is representative of all rods. See page 23.

"representative of all rods" is a rather strong statement, especially given that within any given tooth prism angles change and it is known that within the prism the crystallite packing and alignment varies from DEJ to surface.

The images provided in Figure S2 are not next to the PIC maps, but in the supplemental information, making a direct comparison between morphological c-axis arrangement in crystals

somewhat cumbersome. Furthermore, the additional SEM images are also different scales compared to the PIC maps, that is about twice the field of view for the low magnification and less than half the field of view for the higher magnification images. Nevertheless, the Figure S2 B and D clearly show that crystallite morphological alignment does vary and is not strictly parallel, as the authors claim so emphatically. If the strictly parallel arrangement was so obvious, an SEM image of the same magnification and size of field of view next to the PIC map would make that point. But that information is still missing and hence, the argument is not convincing, as variability in crystallite alignment is clearly visible within both prisms and interprismatic enamel.

We thank reviewers 1 and 2 for stimulating us to do this heroic experiment. It has been incredibly difficult to recognize precisely the same regions with different methods, and this was completely impossible in the middle of the tooth, where the Hunter-Schreger-Bands in Figure 2 were, even though we had the precise coordinates in PEEM, and the precise position recorded in Photoshop during the PEEM experiment on the VLM image. The rods changed too much after etching, and they became unrecognizable. So we resorted to another tooth and another area, where we could, with difficulty, orient ourselves near a cusp of the tooth, and thus could obtain the data the reviewer asked for. The new data are presented in Figure 3 page 5, and discussed on page 4: "Figure 3 shows the same area of enamel imaged in a PIC map before etching and SEM after etching, at precisely the same magnification and in precisely the same location in the tooth. The two images are differently warped by the two microscopes, but they are recognizably from the same region. Their comparison shows that, according to the PIC map, crystals in a rod that are differently oriented by almost 90° (green and magenta pixels in Figure 3D) appear all approximately horizontal and aligned parallel to one another in the SEM image in Figure 3C."

Critique: Importantly, that the structural organization of enamel crystallites into undulating prism and inter-prismatic enamel it is known and very well described for many species and illustrated in many publications. Based on this body of literature one would expect the crystallite axis and alignment between crystallites to change continuously, that is to go in and out of plane of sectioning, a pattern known as Hunter-Schreger-bands, as well as to change their orientation within prisms.

Rebuttal: The reviewer is right, the elongation direction of the rods changes going in and out of plane and thus creating the decussating pattern also known as Hunter-Schreger-Bands (e.g. Figure 2). The new observation here is that within a rod (head) there is a significant degree of misorientation, and neighboring crystals at the nanoscale differ in orientations by 1°-30°. This novel observation challenges the consensus opinion in the field that c-axes of neighboring crystals in enamel rods are co-aligned (see ref. 15: Travis & Glimcher, JCB, 1964, and ref. 14: Glimcher et al. J. Ultrastr. Res., 1965). Importantly the difference in the c-axes orientation are significantly greater than the misorientation of the long axes of the crystals (Figure S1, and new Figure S2 on page 23). We state now much more clearly that the misorientation is within the rods. See summary in page 1, findings on page 3, 1st paragraph, and that this observation cannot be due to the rods changing direction, as the reviewer suggests. See page 5, 2nd paragraph after the figure caption.

The authors do state their interpretations now much more clearly. The PIC maps are impressive and beautiful and maybe there could be clear evidence in Figures S5 and S6 that crystallite long axis and crystallographic c-axis differ. A SEM image simply showing crystallite morphology in the ca. 30 μm aprismatic enamel layer might drive the point home and remove the doubt and counter argument that crystallites within aprismatic enamel are perpendicular to the crown

surface and not necessarily perpendicular to the edge of the section, which may or may not be slightly out of plane.

We greatly thank the reviewer for this suggestion. We have etched the same sample analyzed in PEEM for PIC mapping, and obtained SEM images of the elongation direction of all nanocrystals in aprismatic enamel. We now present these new data in Figure S14, page 36, and comment about them on page 10-11: "After etching, in the same region of aprismatic enamel presented in Figure 6, the crystals indeed appear perpendicular to the tooth surface, as previously observed²⁷ and as shown in Figure S14, but their *c*-axes are almost parallel – not perpendicular – to the tooth surface. This is not the general orientation, in fact in the aprismatic enamel in Figure S13AB we see the *c*-axis $\sim 30^\circ$ from the normal to the tooth surface, $\sim 0^\circ$ from the normal in Figure S13DEFGHI, and $\sim 66^\circ$ from the normal in Figures 6 and S14. The orientation, therefore, appears to be completely uncorrelated with the surface orientation."

Critique:

....although it is known that dehydration and loss of organic material affects the mechanical properties of tooth enamel (Baldassarri et al 2008).

Rebuttal: We have previously demonstrated that dehydrated and later rehydrated enamel has the same mechanical properties as enamel which was never dehydrated (Baldassarri et al. 2008, now cited as ref. 66, discussed on page 8, 1st paragraph).

The rebuttal argument taken from Baldassarri et al. refers to the sample processing procedure and the point that a freeze-dried sample can be rehydrated and that mechanical properties are the same before freeze drying and after re-hydrating. Whereas all the experimental data of the Baldassarri et al. reference show that samples without water or without organic material have very different properties compared to samples that are wet or with organic material. These data and this argument should be considered for the MD modeling the neglects both organic material and a hydration layer between crystallites.

We agree with the reviewer that, as many other materials do, enamel exhibits different mechanical properties dry and wet, with and without organics. We respectfully disagree, however, that water or organics should be considered in MD simulations. The MD simulations results will be completely different if we introduced water layers or organic layers. It is already well-known, and extremely well documented in the literature that at materials discontinuities, such as organic or water layers, crack deflection occurs. See the example of nacre, silica spicules from sponges, sea urchin teeth, etc. It is not interesting to repeat this well-known fact. The interesting new discovery here is that slight mis-orientation alone is a new toughening mechanism. We clarified this point in the discussion on page 7: "as it is well known that at such heterogeneous materials interfaces cracks are normally deflected^{47,56,57}."

Critique:

- The modeling approach and data interpretation needs not acknowledge that the presence of lattice defects and substitutions can affect the crystallite structure.

Rebuttal: We agree and would be glad to remove it, but we cannot find where this was mentioned.

The rebuttal is, unfortunately, focusing on the typo of "not" which would correctly be "needs _to_" acknowledge...

I apologize for the typo, but because this is a very critical point it _does need to be addressed_ in the manuscript.

- The modeling approach and data interpretation needs to acknowledge that the presence of lattice defects and substitutions can affect the crystallite structure.

Thank you for this clarification. We have explicitly acknowledged this on page 7: "The presence of apatite lattice defects and substitutions would affect the crystallite structure and therefore the mechanical response of crystals in MD simulations. These were omitted to keep the model as simple and informative as possible."

Overall, the paper shows truly beautiful PIC maps and electron microscopy data

Thank you!

but leaves the reader wondering why the authors are forcing the point of discerning between morphological long axis and crystallographic c-axis, when the parsimonious explanation would be that the controlled mismatch in crystal long axis alignment optimizes the mechanical properties of enamel.

Yes, that would be the parsimonious explanation, but it is not what is observed in the new Figure 3, and in Figure S2. We sincerely thank both reviewers 1 and 2, for asking us to do the difficult experiment in Figure 3. The paper is much better for it, thus the reviewers have been extremely helpful.

We appreciate the reviewer's time and efforts.

Reviewer #2 (Remarks to the Author):

The authors have provided an extensive rebuttal to the initial review. References are now updated and the list was extended appropriately. Overall however there were many additions to the supplemental portion of this study, but little change to the original material or data as provided in the first submission was made.

As mentioned in the critiques of the first review, a single enamel rod shows a change in fiber orientation from a more parallel orientation of the fibers' long axes to the rod axes to an angle of about 30 degree as laid out by Meckel and others in the 1960. The authors responded to this comment with: "What we are focusing on is the crystallographic misalignment in each rod (head), which contradicts the previous paradigm, mostly based on SAED diffraction studies." This is not true. Meckel's schematic shows exactly this and change in orientation with a single rod!

The reviewer is right, Meckel and others in 1960s developed the fan model of enamel, in which the majority of the crystals in enamel rod are not parallel to each other but fan out as shown in the schematic pasted here from Habelitz 2001 (ref. 22). With advances in microscopy techniques a more nuanced picture has emerged. Numerous TEM studies (some of which are referenced in the manuscript) clearly demonstrated that crystals in the rod are co-oriented (this is a biased measurement, as we explain the manuscript). Later SEM and AFM studies showed that the crystals in the rod appear co-oriented with each other and with the long axis of the rod. For a quicker inspection we paste below three relevant figures for this conversation: the obsolete model by Meckel, and two AFM results showing parallel crystals in the rods: Figure 2 in Habelitz 2001 and Fig. 1 in Uskoković 2008. Lubarsky 2012 also shows similar results. All 3 are cited in the manuscript. We now more clearly state these structural observations as well-established in the introduction. See page 2: "The elongated crystals in each rod run parallel to one another ²²⁻²⁴," and "Crystal elongation direction varies gradually from the rod to the interrod ²⁵⁻²⁷."

orientations in the head and tail area. Inter-rod enamel is enriched in organic matter. Modified from Meckel et al. (1965).

FIG. 1 in Uskoković et al. 2008 ref. 24. Atomic force microscopy images of the microstructure of human enamel: (a) Keyhole-shaped enamel rods of about 5 μm diameter run from the dentin–enamel junction to the surface of the tooth and (b) comprise aligned apatite crystals 40 to 60 nm wide and several hundreds of micrometers long.

In addition, the authors continued to state: “This means that, contrary to earlier reports, the long axis of each nanocrystal is not necessarily co-aligned with the crystalline c-axis, they can be as much as 60° apart.” There seems to be a misconception: Apparently SEM, TEM and AFM all show that fibers flare laterally towards the tail of a key-hole shaped enamel rod. Therefore the c-axis of these fibers will also flare, which is what is shown in the PIC maps of Fig 1.

Indeed. We see this flaring from head to tail in each rod-interror, we are 100% in agreement here. We show heads (H) and tails (T), and their mis-orientations from blue to cyan to green in Figure 1.

What Figures 1 and 2 do *not* show is a radial flaring or fanning of orientations within the rod or head, flaring from its center, as show at the top face of the schematic above. If there were such flaring the rods imaged perpendicular to the image plane should show orientations changing from the center to the sides of the rod. They do not. The most circular rods in Figure 2 are black on the left red at the center, and greenish on the right. Thus the orientations do not flare from the center of the rod. Instead we see slight misalignment of adjacent crystals with no particular pattern reproduced across all rods.

We now more clearly explain on page 6: “The elongated nanocrystals within each rod are parallel to one another morphologically, as shown before by SEM and AFM²²⁻²⁴ and by the present SEM data in Figures 3, S1 and S2. Their crystallographic orientations, however, vary dramatically, up to 90° across a rod (head). This means that the c-axis in some cases can be perpendicular to the elongation direction of the nanocrystals.”

It is not clear why the authors need to make a distinction and claim that the crystal long axis does not correspond to the crystallographic c-axis.

If in any case this was supposedly true, a full correlation of SEM image and the identical PIC-Map needs to be presented. Therefore the main criticism remained, there is no data provided that shows a clear relationship between the microstructure of enamel, e.g. nanofiber orientation shown by SEM (or TEM) and the crystallographic orientation suggested by PIC mapping. Correlated images need to be provided to support the hypothesis that the crystal long axes does not correspond with the crystallographic c-axis. The PIC map of figure 2 is impressive but needs support from EM data. The main observation of a continuous phase of apatite crystals in the interrod enamel remains and is very interesting and a clear proof of models that others have

suggested without such strong evidence, mainly relying on EM evidence. PIC maps facilitate a more extended view of the interrod orientation.

We thank reviewers 1 and 2 for stimulating us to do this heroic experiment. It has been incredibly difficult to recognize precisely the same regions with different methods, and this was completely impossible in the middle of the tooth, where the Hunter-Schreger-Bands in Figure 2 were, even though we had the precise coordinates in PEEM, and the precise position recorded in Photoshop during the PEEM experiment on the VLM image. The rods changed too much after etching, and they became unrecognizable. So we resorted to another tooth and another area, where we could, with difficulty, orient ourselves near a cusp of the tooth, and thus could obtain the data both reviewers asked for. The new data are presented in Figure 3 page 5, and discussed on page 4: "Figure 3 shows the same area of enamel imaged in a PIC map before etching and SEM after etching, at precisely the same magnification and in precisely the same location in the tooth. The two images are differently warped by the two microscopes, but they are recognizably from the same region. Their comparison shows that, according to the PIC map, crystals in a rod that are differently oriented by almost 90° (green and magenta pixels in Figure 3D) appear all approximately horizontal and parallel to one another in the SEM image in Figure 3C."

However extended crystal orientation maps using SEM and TEM correlated with EBSD were recent shown by Koblischka-Veneva et al., Nano Research 2018.

We have now cited this beautiful paper as ref. 58, even though we don't fully understand how to extract quantitative information on how adjacent crystals are oriented with respect to one another. Since all adjacent pixels are successfully indexed in EBSD maps, we wrote on page 7: "Koblischka-Veneva et al. did not observe any non-apatite material at grain boundaries in their electron backscatter diffraction (EBSD) of enamel, corroborating our interpretation that such materials are rare in enamel⁵⁸."

Furthermore, MD simulation continues to lack a support by any data that cracks would penetrate enamel in the fashion described.

We agree with the reviewer that such data would be extremely desirable, but they are impossible to obtain, because the crystals are too small, and no nanoindentation tip can probe at the 50-nm scale of intra-rod enamel crystals. We recently submitted a proposal for a 3-year NSF grant to do mechanical testing across boundaries of mis-oriented but space-filling adjacent crystals, and that will be done on much larger crystals, in coral skeletons. For now, we have to rely on the beautiful MD simulations to prove this point. We also provide *a-posteriori* evidence in Figure 5. If there was an advantage in slight mis-orientation, tooth enamel would have probably evolved to adopt it. The histogram in Figure 5 shows a peak at 1° and a footprint of 30°, corroborating the idea that slight mis-orientation is better at crack deflection.

We appreciate the reviewer's time and efforts.

Reviewer #3 (Remarks to the Author):

The authors have gone to considerable lengths to both answer and take on board queries and suggestions made by the reviewers.

Additional material and amendments have been added in this respect.

The paper can be published.

Thank you!

We appreciate the reviewer's time and efforts and the positive recommendation.

Editorial Note: Reviewer #2 was unable to provide a new review, so a new Reviewer #4 was asked to assess the authors responses and comment on the revision.

Reviewers' comments:

Reviewer #1 (Remarks to the Author):

Thank you for going the extra mile to add SEM data for comparison. The SEM data and their integration into the figures showing PIC data is extremely helpful. The data highlight the coherency of interprismatic enamel, especially taken together with the evidence presented in figure 6 and figure S14. Together these data clearly show the coherency of interprismatic enamel and aprismatic enamel extending to the crown surface, as reflected in the PIC data for aprismatic surface enamel. This is impressive and very interesting!

SEM figure S14 E with arrow labelling as next to the PIC data of the same area clearly support the point the authors are trying to make in the paper and, therefore, should not be in the supplemental information but integrated with figure 6 in the main body of the manuscript.

Figure 4 is completely consistent with the findings on inter/aprismatic enamel that is beautifully demonstrated in the comparison between SEM and PIC data. However, the prismatic enamel does not follow the same pattern of discrepancy between crystal shape long axis and PIC-determined crystallographic c-axis seen in the inter-/aprismatic enamel. In fact, this in itself is very interesting and the main point of the paper that opens up new ways to conceptualize enamel formation. This in itself is new and a great finding. It is unfortunate that the authors force the new insights gained on the discrepancy between morphological long axis and crystallographic c-axis seen in aprismatic enamel onto prismatic enamel. The comparison between SEM and PIC data of the given tooth location in figure 3 clearly shows continuity of interprismatic enamel, but also demonstrates that in prismatic enamel the morphological long axis of the crystallites is congruent with the crystallographic c-axis. There is no need to present modeling data that neglect well-known facts (such as presence of water) in order to extend the new findings to all enamel, in particular, prismatic enamel where it is not supported by the presented data. It is this claim that diminishes the stunning findings and validity of the paper.

Reviewer #4 (Remarks to the Author):

I evaluated the 2nd rebuttal of the authors in response to reviewer #2 and whether there were any other major issues in the manuscript report.

General Comments:

The main outstanding issue that needed to be resolved from the first round of review was the notion that the crystallographic c-axis of enamel apatite crystals within one rod in some areas are not aligned with the crystal long axis. They can deviate from 30 degrees to up to 90 degrees. This is a novel and unusual finding given the amount of literature we have on the structure and morphology of apatite crystals being elongated along the c-axis.

The authors provide both SEM and TEM images with accompanying electron diffraction of crystals in the rod. The new PIC images on aprismatic enamel are further evidence that the elongated crystals are not oriented with their c-axial crystallographic axis in the aprismatic enamel but with an angle.

While the SEM images show bundles of elongated nanocrystals of the areas of interest, the only way to prove the orientation of crystallographic c-axis with regard to the shape of the crystals is SAED of the crystals with given morphology.

The authors provide one TEM image (Fig S6) with corresponding SAED that provide the smoking gun for the presence of adjacent apatite crystals (D1 and D2) not being oriented with their c-

crystallographic axis parallel to each other. This is a a very difficult image for interpretation but convincing. Unfortunately, it is still hard to see two morphologically long crystals in D.

Specific comments:

-I believe this is a very critical image that could be moved to the main paper (if space allows) and not in the SM. I recommend to give a hint as where D1 and D2 can be located in the image? Can they be identified based on the d-spacing (lattice dimensions) measurement on the image??

-I recommend that the authors clarify that this gradual change of crystallographic axis from Zero to 90 does/ or does not happen throughout the entire areas of enamel but only (maybe) selected areas in the bulk of enamel. Maybe areas that will withstand the chewing forces (like the occlusion surfaces or cusps of enamel).

How many teeth the authors examined??

Maybe the word "in some cases" need to be added to the abstract sentence when the gradual 90 degree mis orientation is considered. Note that human molars can be pretty heterogeneous in their micro structure depending which region is examined.

- Does the 90 degree deviation from c-axis mean that the crystals are aligned with their a or b crystallographic axis?? Please clarify in the discussion. Does this mean that apatite crystals with new "morphology" are formed? Elongated along a or b instead of c??

It is a bit difficult to imagine that a an hexagonal crystal system such as fluoridated carbonated apatite in enamel can elongate itself 90 degrees from the C-axis and an extraordinary control over this kind of growth would be needed in the enamel extracellular matrix.

- In their recent JACS paper the authors do not report such "misorientation" of crystals within the rods in the case of mice incisor, or maybe they did not look close enough. Comparing these findings regarding human and mice is worthy of discussion at the end of this paper.

-How the pixels in Fig S12 correspond to distance in nm?? It would be useful to add it to the figure so to give an idea of resolution of PIC when compared to TEM.

In summary: The main concerns of reviewer #2 have been adequately addressed and the paper is now suitable for acceptance after addressing the above minor comments.

Janet Oldak

Reviewers' comments:

Reviewer #1 (Remarks to the Author):

Thank you for going the extra mile to add SEM data for comparison. The SEM data and their integration into the figures showing PIC data is extremely helpful. The data highlight the coherency of interprismatic enamel, especially taken together with the evidence presented in figure 6 and figure S14. Together these data clearly show the coherency of interprismatic enamel and aprismatic enamel extending to the crown surface, as reflected in the PIC data for aprismatic surface enamel. This is impressive and very interesting!

Thank you

SEM figure S14 E with arrow labelling as next to the PIC data of the same area clearly support the point the authors are trying to make in the paper and, therefore, should not be in the supplemental information but integrated with figure 6 in the main body of the manuscript.

Thanks for this suggestion. The previous Fig. S14 is now Fig. 7. **Page 12.**

Figure 4 is completely consistent with the findings on inter/aprismatic enamel that is beautifully demonstrated in the comparison between SEM and PIC data. However, the prismatic enamel does not follow the same pattern of discrepancy between crystal shape long axis and PIC-determined crystallographic c-axis seen in the inter-/aprismatic enamel. In fact, this in itself is very interesting and the main point of the paper that opens up new ways to conceptualize enamel formation. This in itself is new and a great finding.

We agree, as was already described in the paper.

It is unfortunate that the authors force the new insights gained on the discrepancy between morphological long axis and crystallographic c-axis seen in aprismatic enamel onto prismatic enamel. The comparison between SEM and PIC data of the given tooth location in figure 3 clearly shows continuity of interprismatic enamel, but also demonstrates that in prismatic enamel the morphological long axis of the crystallites is congruent with the crystallographic c-axis.

Did the reviewer mean “incongruent” rather than “congruent”? If he/she really meant “congruent” we must disagree. In Fig. 3B inter-prismatic enamel indeed shows one dominant color, green = 30° c-axes, but the prismatic enamel shows all the colors of the spectrum within a simple prism (or rod, as we call them in the paper). This corresponds to a change in c-axis orientation from -90° to +90°. The SEM image of precisely the same prism in Fig. 3A shows that all crystals within that prism are elongated parallel to one another like raw spaghetti. In the small region of a single rod highlighted in Fig. 3C and D, the SEM image shows that the crystals are elongated horizontally, but the PIC map shows that the c-axes of the crystals in this same region range from red near the top to yellow in the middle and magenta at the bottom of the image (±90°, +60°, and -60°, respectively). While it may be argued that the SEM image shows that the crystals aren't *exactly* parallel to each other, the very small deviations in the crystal alignment is much smaller than the large (>30°) c-axis misorientations shown in the PIC map. Based on the higher magnification images in Fig. 3CD, it is clear that the PIC maps indicate a large angular spread in c-axis orientations that does not correspond to a similar change in elongation direction in the SEM images. Perhaps the reviewer could be more specific in where he/she sees the “congruence”?

There is no need to present modeling data that neglect well-known facts (such as presence of water) in order to extend the new findings to all enamel, in particular, prismatic enamel where it is not supported by the presented data. It is this claim that diminishes the stunning findings and validity of the paper.

We could remove the simulation results, and the increased toughness of enamel provided by small mis-orientations of adjacent crystals. But, isn't it better to discover a structure *and its function*, than simply describe the structure?

Regarding the fact that there is water and organics in enamel, these are indeed well-established facts. We note that water tends to co-localize with organics, and that at interrod interfaces there are organic sheaths. Our model, simulations, and data show that mis-orientation is small at the nanoscale, and larger at the microscale, which is typical of hierarchical materials such as enamel. The reviewer refers to a toughening mechanism at the microscale, whereas the focus of this paper is one level down in hierarchy.

We now write on page 10:

“Remarkably, all crystals in human enamel rods are slightly mis-oriented with respect to their neighboring crystals, as shown by the histogram in Figure 6. The few greatly mis-oriented adjacent pixels, e.g. 60°, occur at the rod-interrod boundaries where most mis-oriented crystals are separated by an organic sheath (non-polarization-dependent and therefore black in PIC maps) and only a few are not, generating the small spikes in the histograms of Figures 6 and S11. The model and simulations presented here predict that at greatly mis-oriented grain boundaries no crack deflection occurs, thus crack-deflecting organic sheaths^{47,56,57} located at rod-interrod boundaries are necessary to toughen the material.”

Reviewer #4 (Remarks to the Author):

I evaluated the 2nd rebuttal of the authors in response to reviewer #2 and whether there were any other major issues in the manuscript report.

General Comments:

The main outstanding issue that needed to be resolved from the first round of review was the notion that the crystallographic c—axis of enamel apatite crystals within one rod in some areas are not aligned with the crystal long axis. They can deviate from 30 degrees to up to 90 degrees.

This is a novel and unusual finding given the amount of literature we have on the structure and morphology of apatite crystals being elongated along the c-axis.

Thank you for understanding the paper, and recognizing the novelty of the mis-orientations within each rod. This is in direct contradiction of Reviewer #1's statement above.

The authors provide both SEM and TEM images with accompanying electron diffraction of crystals in the rod. The new PIC images on aprismatic enamel are further evidence that the elongated crystals are not oriented with their c-axial crystallographic axis in the aprismatic enamel but with an angle.

Thanks!

While the SEM images show bundles of elongated nanocrystals of the areas of interest, the only way to prove the orientation of crystallographic c-axis with regard to the shape of the crystals is SAED of the crystals with given morphology.

The authors provide one TEM image (Fig S6) with corresponding SAED that provide the smoking gun for the presence of adjacent apatite crystals (D1 and D2) not being oriented with their c- crystallographic axis parallel to each other. This is a very difficult image for interpretation, but convincing.

Thank you!

Unfortunately, it is still hard to see two morphologically long crystals in D.

We agree. The reason this figure is so hard to interpret is that the FIB'ed section is 100 nm thick, thus multiple crystals fit into it, and overlap. We have attempted to improve the presentation of this figure, as described below.

Specific comments:

-I believe this is a very critical image that could be moved to the main paper (if space allows) and not in the SM.

The previous Fig. S6 is now Fig. 4 in the main text. Page 6.

I recommend to give a hint as where D1 and D2 can be located in the image? Can they be identified based on the d-spacing (lattice dimensions) measurement on the image??

We tried multiple different approaches but none of them improved clarity, in fact they mostly complicated the figure and could not show the outlines of the 3 crystals. We have therefore not changed the figure. We simplified the caption, though, to improve clarity. We moved details of the FIB preparation into the methods page 17, and removed a lot of repetitions from Fig. 4 caption. Page 6-7.

-I recommend that the authors clarify that this gradual change of crystallographic axis from Zero to 90 does/ or does not happen throughout the entire areas of enamel but only (maybe) selected areas in the bulk of enamel. Maybe areas that will withhold the chewing forces (like the occlusion surfaces or cusps of enamel).

We looked all over, and never observed zero angle spread within a rod. As shown in Figures S3 and S5, all of the areas analyzed show color changes within each rod. We now pointed this out more clearly in the summary, and in the text. See page 1 summary, page 7, page 14 conclusions.

How many teeth the authors examined??

Two 3rd molars from young adults, as shown in Figures S3 and S5. This is now more explicitly stated at the beginning of the methods. Page 14. We have analyzed 3 additional 3rd molars after the submission of this paper, with consistent results.

Maybe the word "in some cases" need to be added to the abstract sentence when the gradual 90

degree mis orientation is considered. Note that human molars can be pretty heterogeneous in their micro structure depending which region is examined.

Indeed they are structurally heterogeneous, but we have analyzed all sorts of regions, as summarized in Figures S3 and S5 and find consistent results, as now more explicitly stated in the summary, the main text, and the conclusions. See page 1, page 7, page 14. On page 14 we write “Crystals in the aprismatic and interrod enamel are highly co-oriented across the entire thickness of the enamel layer, whereas in the rods they are mis-oriented slightly (0° - 30°) with respect to their immediately neighboring crystals, and greatly (30° - 90°) across the rod in any orientation. The angle spread within a rod was never observed to be zero.”

- Does the 90 degree deviation from c-axis mean that the crystals are aligned with their a or b crystallographic axis?? Please clarify in the discussion. Does this mean that apatite crystals with new “morphology” are formed? Elongated along a or b instead of c??

It is a bit difficult to imagine that a hexagonal crystal system such as fluoridated carbonated apatite in enamel can elongate itself 90 degrees from the C-axis and an extraordinary control over this kind of growth would be needed in the enamel extracellular matrix.

We don't think that crystal orientation is controlled at all, but of course we do not know for a fact. We now write in the discussion: “The mis-match of c-axis and elongation direction was observed in all crystals, within rods, interrod, and aprismatic enamel, sometimes by as much as 90° . The latter case does not mean that crystals grow along the a- or the b-axis direction. It appears that the crystal orientation is uncorrelated with elongation direction, thus crystals can be oriented in any direction as they grow. This is consistent with two opposing formation mechanisms: (i) crystal growth via an amorphous calcium phosphate precursor phase⁸⁰, with the crystalline phase propagating through and at the expense of the amorphous phase, or (ii) crystal growth by imperfect oriented attachment of previously crystalline nanoparticles⁷¹.” See paragraph split between pages 13-14.

- In their recent JACS paper the authors do not report such “misorientation” of crystals within the rods in the case of mice incisor, or maybe they did not look close enough. Comparing these findings regarding human and mice is worthy of discussion at the end of this paper.

Excellent suggestion. We have now compared the two. Second paragraph of page 14.

-How the pixels in Fig S12 correspond to distance in nm?? It would be useful to add it to the figure so to give an idea of resolution of PIC when compared to TEM.

We have added the pixel size to Figure 6 and S11, page 9 and 36.

In summary: The main concerns of reviewer #2 have been adequately addressed and the paper is now suitable for acceptance after addressing the above minor comments.

Janet Oldak

Thank you!

1. Killian CE, *et al.* (2011) Self-sharpening mechanism of the sea urchin tooth. *Adv Funct Mater* 21:682–690.
2. Miserez A, *et al.* (2008) Effects of laminate architecture on fracture resistance of sponge biosilica: Lessons from nature. *Adv Funct Mater* 18(8):1241-1248.
3. Aizenberg J, *et al.* (2005) Skeleton of *Euplectella* sp.: structural hierarchy from the nanoscale to the macroscale. *Science* 309(5732):275-278.

REVIEWERS' COMMENTS:

Reviewer #1 (Remarks to the Author):

Thank you for going the extra mile to add SEM data for comparison. The SEM data and their integration into the figures showing PIC data is extremely helpful. The data highlight the coherency of interprismatic enamel, especially taken together with the evidence presented in figure 6 and figure S14. Together these data clearly show the coherency of interprismatic enamel and aprismatic enamel extending to the crown surface, as reflected in the PIC data for aprismatic surface enamel. This is impressive and very interesting!

Thank you

SEM figure S14 E with arrow labelling as next to the PIC data of the same area clearly support the point the authors are trying to make in the paper and, therefore, should not be in the supplemental information but integrated with figure 6 in the main body of the manuscript.

Thanks for this suggestion. The previous Fig. S14 is now Fig. 7. Page 12.
This is good.

Figure 4 is completely consistent with the findings on inter/aprismatic enamel that is beautifully demonstrated in the comparison between SEM and PIC data. However, the prismatic enamel does not follow the same pattern of discrepancy between crystal shape long axis and PIC-determined crystallographic c-axis seen in the inter-/aprismatic enamel. In fact, this in itself is very interesting and the main point of the paper that opens up new ways to conceptualize enamel formation. This in itself is new and a great finding.

We agree, as was already described in the paper.

It is unfortunate that the authors force the new insights gained on the discrepancy between morphological long axis and crystallographic c-axis seen in aprismatic enamel onto prismatic enamel. The comparison between SEM and PIC data of the given tooth location in figure 3 clearly shows continuity of interprismatic enamel, but also demonstrates that in prismatic enamel the morphological long axis of the crystallites is congruent with the crystallographic c-axis.

Did the reviewer mean "incongruent" rather than "congruent"? If he/she really meant "congruent" we must disagree. In Fig. 3B inter-prismatic enamel indeed shows one dominant color, green = 30° c-axes, but the prismatic enamel shows all the colors of the spectrum within a simple prism (or rod, as we call them in the paper). This corresponds to a change in c-axis orientation from -90° to $+90^\circ$. The SEM image of precisely the same prism in Fig. 3A shows that all crystals within that prism are elongated parallel to one another like raw spaghetti. In the small region of a single rod highlighted in Fig. 3C and D, the SEM image shows that the crystals are elongated horizontally, but the PIC map shows that the c-axes of the crystals in this same region range from red near the top to yellow in the middle and magenta at the bottom of the image ($\pm 90^\circ$, $+60^\circ$, and -60° , respectively). While it may be argued that the SEM image shows that the crystals aren't exactly parallel to each other, the very small deviations in the crystal alignment is much smaller than the large ($>30^\circ$) c-axis misorientations shown in the PIC map. Based on the higher magnification images in Fig. 3CD, it is clear that the PIC maps indicate a large angular spread in c-axis orientations that does not correspond to a similar change in elongation direction in the SEM images. Perhaps the reviewer could be more specific in where he/she sees the "congruence"?

It seems that this difference in seeing and interpreting how PIC data and SEM data relate to each other can't be quite resolved, which is acceptable. More importantly, the following part of the authors' response is critical and makes a big difference for this very conflict and the data interpretation and

should be part of the discussion in the paper as it presents the data with some slight in emphasis to highlight the difference between the SEM impression of crystal shape and the PIC result on c-axis orientation:

"While it may be argued that the SEM image shows that the crystals aren't exactly parallel to each other, the very small deviations in the crystal alignment is much smaller than the large ($>30^\circ$) c-axis misorientations shown in the PIC map. Based on the higher magnification images in Fig. 3CD, it is clear that the PIC maps indicate a large angular spread in c-axis orientations that does not correspond to a similar change in elongation direction in the SEM images."

There is no need to present modeling data that neglect well-known facts (such as presence of water) in order to extend the new findings to all enamel, in particular, prismatic enamel where it is not supported by the presented data. It is this claim that diminishes the stunning findings and validity of the paper.

We could remove the simulation results, and the increased toughness of enamel provided by small mis-orientations of adjacent crystals. But, isn't it better to discover a structure and its function, than simply describe the structure?

Regarding the fact that there is water and organics in enamel, these are indeed well-established facts. We note that water tends to co-localize with organics, and that at interrod interfaces there are organic sheaths. Our model, simulations, and data show that mis-orientation is small at the nanoscale, and larger at the microscale, which is typical of hierarchical materials such as enamel. The reviewer refers to a toughening mechanism at the microscale, whereas the focus of this paper is one level down in hierarchy.

We now write on page 10:

"Remarkably, all crystals in human enamel rods are slightly mis-oriented with respect to their neighboring crystals, as shown by the histogram in Figure 6. The few greatly mis-oriented adjacent pixels, e.g. 60° , occur at the rod-interrod boundaries where most mis-oriented crystals are separated by an organic sheath (non-polarization-dependent and therefore black in PIC maps) and only a few are not, generating the small spikes in the histograms of Figures 6 and S11. The model and simulations presented here predict that at greatly mis-oriented grain boundaries no crack deflection occurs, thus crack-deflecting organic sheaths 47,56,57 located at rod-interrod boundaries are necessary to toughen the material."

The addition of this paragraph is helpful and as it states that "...all crystals in human enamel rods are slightly mis-oriented ...The few greatly mis-oriented adjacent pixels, e.g. 60° , occur at the rod-interrod boundaries where most mis-oriented crystals are separated by an organic sheath (non-polarization-dependent and therefore black in PIC maps) and only a few are not." This presents and integrates the modeling data better. While the argument of hierarchical organization is correct, and organic material is located in mature enamel as prism sheath mostly at the interface rod-interrod interfaces, water is still also between crystallites within the rod." The addition of the rebuttal argument "Our model, simulations, and data show that mis-orientation is small at the nanoscale, and larger at the microscale" would also help to leave more room for interpretation.

Overall, the authors have made very helpful adjustments to the manuscript, including rearrangement of figures and addition of text, to address the review comments and provide additional clarity. This does change the tone to leave some room for a difference in opinion and data interpretation, which is certainly acceptable.

The crux of the paper lies in the critical validation of the claims through TEM analyses. While the addition of TEM images with SAED into the main text is great, it does not resolve the problem that a) it is extremely difficult technically to prepare a sample capturing only one crystal layer and b) it is

extremely difficult to discern the different crystallites and hence the morphological crystal long axis. While the argument of the paper hinges on these data, they might be convincing for some but not for others. This does leave an unresolved weakness and creates some agony. However, it is clear that at this time there is no straightforward technical and analytical approach to resolve this issue, remove all doubt, and silence the annoying critics. Scraping enamel with a fine needle to break out crystallites is used by some as a way to sample. Some very few crystallites might separate and create an almost literal search for the needle in the haystack, since this might be a way (although rather painful) to obtain TEM data where single crystallites can be discerned. However, integrating the discussion on these difficulties into the main text would add to the strengths of the paper and certainly encourage discussion in the field and follow up studies.